

# Numerical airfoil catalogue including 360° airfoil polars and aeroacoustic footprints

Manfred Imiela[1], Benjamin Faßmann[1], Gerrit Heilers[1], Gunther Wilke[1]

[1]Institute of Aerodynamics and Flow Technology, DLR, Braunschweig, 38102, Germany

*Correspondence to*: Manfred Imiela (Manfred.imiela@dlr.de) or Benjamin Faßmann (Benjamin.Fassmann@dlr.de)

**Abstract.** A methodology is presented for generating 360° airfoil polars and aeroacoustic characteristics by means of CFD and CAA. The aerodynamic procedure is validated against experimental data of the well-known airfoils DU-93-W-210 and DU-97-W-300. While a better prediction of the aerodynamic coefficients in the range of -30° and 30° is achieved by a combination of the k-ω SST turbulence model and a C-topology mesh, for the remaining angles of attack more confidence is
gained with the SA negative turbulence model in combination with an O-topology mesh. Therefore the two data sets are subsequently fused to one complete data set using a kriging interpolation approach. The result of ten different airfoils using the proposed method is presented. For providing the aeroacoustic characteristics for a wide operation range four computations and a bilinear interpolation are needed, since the aeroacoustic is dependent on the Mach and Reynolds number. The bilinear interpolation approach is verificated by a comparison between the originally simulated and the emulated result
at a fifth computational set for six different airfoils. The corresponding overall sound pressure level (OASPL) for four angles of attack for these airfoils is presented and the difference between a fully turbulent computation and simulations with fixed transition is assessed. The aeroacoustic results further include high-fidelity directivity functions.

## 1 Introduction

The present work is carried out in the scope of the DLR project RoDeO. The goal of the project is the design of a real wind
turbine rotor. While a proper evaluation of the aerodynamics and a sound structural dimensioning is mandatory, the aeroacoustic analysis is an additional goal of the project in order to design a quiet rotor. Therefore the present paper is focussed on the generation of airfoil polars including aerodynamic coefficients and aeroacoustic characteristics.

During the design process of a wind turbine blade a multitude of load cases has to be considered. Consequently an engineering model is needed that can handle the amount of computations required. Despite their limitations computer
programs that are based on the Blade Element Method (BEM) method are still widely used for certification and in the first phase of the design process (Snel 2003, Ning 2013) in order to determine the aerodynamic loads. These codes require the aerodynamic polar tables containing the aerodynamic lift ($C_l$), drag ($C_d$) and moment coefficient ($C_m$) as input. In contrast to polar tables used for aeronautical applications, aerodynamic coefficients for the use in wind energy applications need to be provided for the complete range of angles of attack (from 0° to 360°). The generation of such polar tables is by no means





trivial. Although research has been ongoing for decades the physical phenomena involved – namely accurate stall prediction and rotational and 3D effects - have not yet completely been understood (Tangler 2005). While quite some progress has been made in recent years on understanding the phenomenon of rotational augmentation (Gross 2012, Bangga 2016) the methods adopted in the said publications are still too expensive in order to be applied on a vast number of airfoils. In contrast to the

very expensive methods a manifold number of approximation methods exit for completion of measured or computed data sets in the regular range of angles of attack. These approximation methods are mostly based on a combination of harmonic functions and require certain empirical input parameters. In most cases measured data is obtained by wind tunnel testing and subsequently completed and corrected by such methods. The reason for their popularity lies in the easy handling and the inexpensive evaluation. On the other hand these methods suffer some severe drawbacks, e.g. they "seem incapable of

representing different maximum $C_l$ values in the negative and positive α region" and "they do not seem to capture the peculiarities of the polar in or close to the attached flow region as well as around 180°" (Skrzypiński 2014).

The authors are conscious of the fact that with the application of steady 2D RANS simulations for the complete range of angles of attack they leave the boundary of validity of this approach. Yet they believe that the results from a steady RANS simulation will be superior to the results of the commonly applied approximation methods. Since the computation of a

complete polar by means of steady 2D RANS computations can nowadays be achieved within a day, the chosen prospect can be viewed as a good compromise between accuracy and efficiency. The computed polar tables will be based on pure steady 2D computations without corrections. The corrections for dynamic stall and rotational effects are sought to be included by the comprehensive rotorcraft code that will be used in the RoDeO project.

Wind turbines offer numerous sources of noise. Besides mechanical sources, like the gear box and the generator, there are

aerodynamic (aeroacoustic) sources of sound to be perceived. Aeroacoustic noise dominates the overall sound emission of a wind turbine (Wagner 1996). Numerous sound mechanisms at the rotor blades are differentiated in the literature. These are separation noise, vortex the shedding noise, tip vortex noise, and turbulent boundary-layer trailing-edge noise (TBL-TEN) (Brooks 1989). For multi megawatt turbines, the TBL-TEN has been identified to be the prominent sound contribution (Oerlemans 2007).

For noise prediction, semi empirical tools are widely used, e.g. the NREL NAFNoise (Moriarty 2005). Different sound sources are included in that framework. It is based on the BPM model (Brooks 1989). A model for inflow turbulence noise by Amiet 1975 is implemented as well. Further, the aerodynamic tool XFOIL (Drela 1987) is also enclosed to NAFNoise. The same holds for the TNO model (Parchen 1998). All these models can independently be used to predict emitted noise in the context of wind turbines. In total, they show a good agreement in overall sound pressure levels, but separate validation is

often lacking in detail.

The approach within the framework of RoDeO concentrates on the high fidelity numerical investigation of the emitted sound of the rotor blades, based on 2D computations of TBL-TEN caused by stochastically modelled turbulence. A reduced order model unites the reliability of highly resolved computations with a fast estimation procedure. Validation will be realized for each step of the procedure.



## 2 Tools & Methodology

### 2.1 Framework for Generating Aerodynamic Polars

The framework is depicted in Figure 1. The employed tools are coloured in blue, the inputs/outputs between the tools are coloured in orange and the final result is highlighted in green. In the first step two meshes around the airfoil are generated –

5 one C-topology mesh and one O-topology mesh. In the next step the automatic process between PAPST (Parametric Airfoil Polar Simulation Tool) and TAU is initiated in terms of a master slave process. Consequently the desired computations for all angles of attack and Reynolds number combinations are started automatically and the resulting airfoil coefficients ($C_l$, $C_d$, $C_m$) are returned to the master process. The transition location within each calculation is predicted by two additional programs, namely COCO for the boundary layer analysis and LILO for the stability analysis. An overview of this procedure

can be found in Krumbein 2017. For more detailed information the reader is referred to Krumbein 2009. At the end two different data sets have been generated – one on the C-topology mesh for angles of attack between -30° and 30° and another one on the O-topology mesh for angles of attack between -180° and 180°. Subsequently the two data sets are manually passed to the program POT as developed by Wilke 2012 in order to fuse the two data sets to one complete data set.

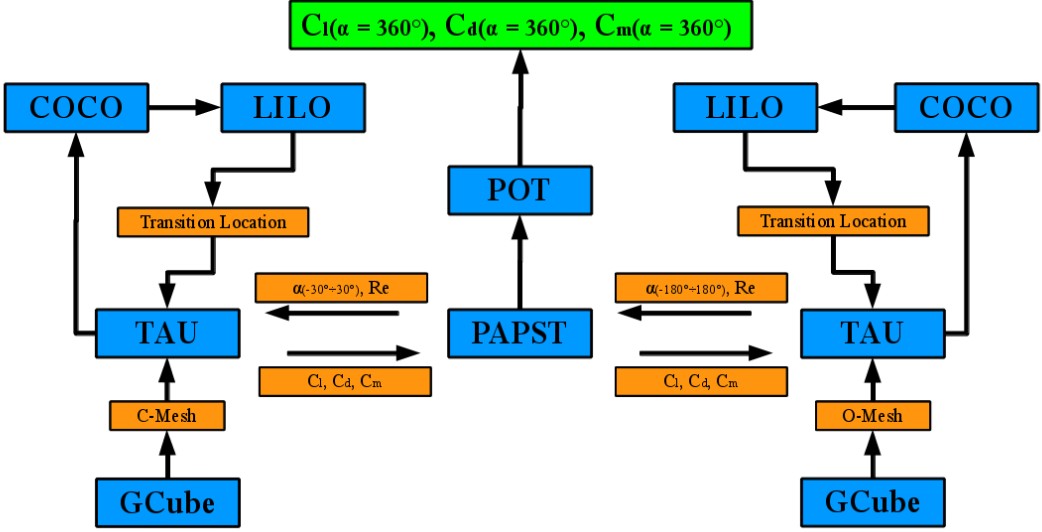

**Figure 1: Aerodynamic Framework for generation of airfoil polar tables**

### 2.2 CFD Solver TAU

The TAU-Code (Schwamborn 2006) developed at the Institute of Aerodynamics and Flow Technology is used for the aerodynamic simulations. It solves the compressible, three-dimensional Reynolds-Averaged Navier-Stokes (RANS) equations using a finite volume formulation. The program consists of several modules that are integrated in the simulation





environment FlowSimulator. This python-based framework performs the coupling between the different modules and allows in-memory data exchange.

The TAU-Code uses a cell-vertex formulation with a dual-grid approach for the spatial discretization. The solver module contains a central scheme as well as several upwind schemes for the discretization of the inviscid fluxes. Viscous terms are
computed with a second-order central scheme. For artificial dissipation scalar or matrix dissipation might be chosen by the user. The unstructured code supports the use of hybrid meshes, i.e. mixed type of elements (hexaeder, tetraeder, etc.) offering more flexibility in the mesh generation process.

Time integration is achieved using either an explicit Runge-Kutta type time-stepping scheme or an implicit LU-SGS (lower-upper symmetric Gauss-Seidel) algorithm. The time-accurate simulations are performed with an implicit dual-time stepping
approach. Various multi-grid type cycles are available for accelerating the convergence of the flow equations.

For the simulation of turbulent flows several one- and two-equation turbulence models are implemented.

**2.3 Computational Aerodynamic Setup**

The first dataset is generated using a C-topology mesh as depicted in Figure 2 (top). The mesh consists of hexahedral elements only and is made up of 251 points in chordwise and 241 points in normal direction. The wake includes an
additional 51 points and is inclined by 5° in order to account for the downwash of the airfoil. In accordance with the Turbulence Modeling Research by NASA 2014 the farfield distance was set to 100 chord lengths in all directions in order to reduce the influence on the flow around the airfoil. Smoothing techniques were applied to ensure that normals are perpendicular to the greatest possible extent. Horizontally the first cell in the wake below the upper trailing edge point is equally sized as the one above. The growth rate was tried to keep as low as possible while not increasing the total amount of
points above acceptable limits. The total amount adds up to approximately 80.000 mesh points.



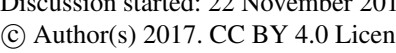


**Figure 2: Various perspectives (complete mesh, complete airfoil, trailing edge close-up) of computational meshes used for generating airfoil polars: top: C-topology, bottom: O-topology**

The second dataset is obtained using an O-topology mesh as shown in Figure 2 (bottom). This mesh consists also of
hexahedral elements only and is made up of approximately 200.000 mesh points. After the mesh generation process the mesh
is also smoothed to ensure orthogonality of normals as much as possible. Due to the high chord wise resolution the mesh
reaches a high smoothness in all areas of the mesh, even at the trailing edge. Due to the O-topology a stronger stretching of
the cells especially towards the trailing edge is inevitable.

Besides the mesh setup the computational setup also covers the numerical settings. This includes settings for parameters
regarding the numerical scheme, parameters for turbulence and transition as well as the fundamental parameters describing
the flow conditions.

The relevant parameters for the solver settings are summarized in Table 1. Due to the subsonic nature of the flow the implicit
Backward Euler Algorithm has been chosen in order to find a steady state solution to the given problem. The inviscid fluxes
are discretized with a central finite difference scheme. Matrix dissipation is used to stabilize the numerical scheme. The
computation is sped up by a 4v multigrid cycle.





The flow conditions are described by the values listed in Table 2. The velocity an airfoil on a wind turbine is exposed to heavily depends on its radial position. Nevertheless the Mach number has only a negligible influence on the aerodynamic coefficients because the flow always stays subsonic. Therefore a medium speed of approximately 40 m/s has been used for all computations. The crucial characteristic number is represented by the Reynolds number. Because the aerodynamic polars

5  will be used for the design of a small size wind turbine, a Reynolds number of 1 million has been chosen. Each dataset is comprised of 31 calculations since a sampling rate of $\Delta\alpha = 2°$ has been used. For the complete range of angles of attack 97 computations are necessary. Between -180° and 170° and between -30° and 30° the sampling rate was set to $\Delta\alpha = 2°$ while in all other regions of the polar the sampling rate was chosen to be $\Delta\alpha = 5°$.

**Table 1: Solver settings**

| Solver | |
|---|---|
| Time integration | Backward Euler |
| Computation type | Steady |
| Inviscid flux discretization | Central |
| Dissipation scheme | Matrix |
| Multigrid Cycle | 4v |

**Table 2: Flow conditions**

| Flow | |
|---|---|
| Mach number | 0.12 |
| Reynolds number (C-mesh) | 1e6 |
| Reynolds number (O-mesh) | 1e6 |
| Angle of attack (C-mesh) | -30° ÷ 30° |
| Angle of attack (O-mesh) | -180° ÷ 180° |

10  Table 3 shows the settings regarding the turbulence model. At the beginning of the project it was sought to use the aerodynamic results in the small angle of attack range as input for the acoustic computations. Therefore the k-ω SST model is used for those computations since the reconstruction of the artificial turbulence is more accurate if a time- **and** length-scale are provided. The prominent characteristic of the Spalart Allmaras turbulence model is its robustness. Therefore it has been selected for the computation of the complete polar. The turbulent intensity for all computations has been set to 0.001.

**Table 3: Turbulence model parameters**

| Turbulence | |
|---|---|
| Model for C-Mesh | k-w SST |
| Model for O-Mesh | SA |
| Turbulent intensity | 0.001 |

**Table 4: Transition model parameters**

| Transition | |
|---|---|
| Type | $e^N$ method |
| N-factor | 9 |
| Instability type | Tollmien-Schlichting, laminar separation |

The fundamental parameters for the transition prediction are summarized in Table 4. As shown in Figure 1 TAU is coupled to the boundary layer code COCO and the linear stability analysis program LILO in order to predict the transition position. The transition criteria selected are Tollmien-Schlichting instabilities and laminar separation. For the critical N-factor a value of 9 has been chosen.





## 2.4 Parametric Airfoil Simulation Tool (PAPST)

For generating the airfoil polars a vast number of calculations have to be carried out. For this purpose a set of python based scripts have been programed by Gerrit Heilers in order to perform all these computations automatically. Although for this paper only one Reynolds and one Mach number have been chosen, PAPST has been programed to conduct complete DOE

computations with the angle of attack, Reynolds number and Mach number being the design parameters of the DOE. The program contains post-processing routines in order to automatically generate field and pressure plots as for example depicted in Figure 11 and Figure 17. The computations can be conducted on a local machine as well as done in this case on a cluster operated by a queueing system. If one or more computations or even the whole DOE crash due to convergence, other software or hardware errors various techniques for restarting the computations are available. One of PAPST's main features

is the detection of non-optimally or poorly converged solutions and its subsequent averaging techniques. For this purpose in case of an oscillating solution the last three periods are determined (at the most over 25000 iterations) as shown in Figure 3. Subsequently the amplitude and the average value are written to the results file. This process is repeated independently for all three coefficients ($C_l$, $C_d$, $C_m$).

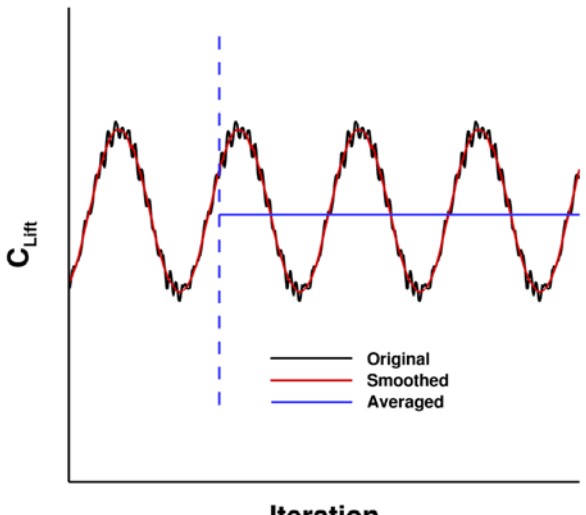

**Figure 3: Averaging technique for obtaining average value for oscillating solution**

## 2.5 Interpolation Model for Data Fusion (POT)

For the data fusion of low- and high-fidelity data, the Powerful Optimization Toolkit (POT) as developed by (Wilke, 2012) has been used. It has been developed for the optimization of helicopter rotor blades including model input from different fidelities. It features surrogate models based on Kriging. For regular (multi-dimensional) interpolation, universal Kriging is

applicable, while for joining functions hierarchical Kriging as postulated by Han and Görtz 2012 is recommended. In order to build a hierarchical Kriging model, first a low-fidelity Kriging model needs to be built:





$$\hat{y}_{lfm}(\vec{\mathrm{x}}) = \hat{\mathrm{f}}_{poly}(\vec{\mathrm{x}}) + \epsilon_{RBF}(\vec{\mathrm{x}})$$

Where $\hat{y}_{lfm}$ is the low-fidelity approximation, $(\vec{\mathrm{x}})$ the vector of independent variables or parameters, $\hat{\mathrm{f}}_{poly}$ a polynomial regression model and $\epsilon_{RBF}$ the error correction to the sample points. For a 1D function, a second order polynomial regression model may be written as:

$$\hat{\mathrm{f}}_{poly}(\mathrm{x}) = \beta_0 1 + \beta_1 x + \beta_2 x^2 = \vec{f} \cdot \vec{\beta}$$

With $\beta_i$ being the coefficients, $\vec{\beta}$ the coefficents vector, and $\vec{f}$ the vector of the polynomials $(1, \mathrm{x}, \mathrm{x}^2)$. Since a regression

model does not necessarily interpolate the data exactly, the offset between regression model and sample points is corrected with radial basis functions:

$$\varepsilon_{RBF}(\mathrm{x}) = \vec{\psi}(\mathrm{x}) \cdot \overline{\Psi^{-1}}(\vec{Y_S} - \overline{F} \cdot \vec{\beta})$$

Here, $\vec{\psi}(\mathrm{x})$ is the correlation vector of the prediction point with the given sample points, $\overline{\Psi}$ the self-correlation matrix of sample points, $\vec{Y_S}$ the vector of sample responses and $\overline{F}$ the matrix of polynomials. The radial basis function employed here is the regular Gaussian-like Kriging basis function:

$$\psi = \exp(-\sum_k \theta_k |x_{i,k} - x_{j,k}|^{p_k})$$

Where k is the spatial direction, $\theta_k$ and $p_k$ are tuning parameters for Kriging. Latter is set to 2 to allow for smooth functions. Even though interpolation is wanted, numerical codes may feature noise from a finite convergence. A common way to deal with this is done by adding a noise constant $\lambda$ to the diagonal of the correlation matrix $\overline{\Psi}$. The values for $\theta_k$, $p_k$ and $\lambda$ may be found through the optimization of the likelihood function. For more details on this and the exact determination of the polynomial coefficients in the Kriging process, the book by Forrester et al. 2008 is recommended.

The variable-fidelity hierarchical Kriging model is now built from this low-fidelity model by exchanging the polynomial trend function with the low-fidelity model and adding a scaling coefficient ρ to it:

$$\hat{y}_{vfm}(\vec{\mathrm{x}}) = \rho \hat{y}_{lfm}(\vec{\mathrm{x}}) + \epsilon_{RBF,hfm}(\vec{\mathrm{x}})$$

The determination of ρ is done in a similar fashion as to $\vec{\beta}$, with the exception that the polynomial vector $\vec{f}$ is now simply the low-fidelity surrogate model $\hat{y}_{lfm}(\vec{\mathrm{x}})$. The error correction is now done in relation to the responses of the high-fidelity model (hfm) at which also a low-fidelity prediction is made with the low-fidelity surrogate model. The correlation matrix is thus

also built from the high-fidelity sample locations. The same commodities as for the universal Kriging are available, therefore the hyper parameters $\theta_k$, $p_k$ (also set to 2 for the high-fidelity) and $\lambda$ may also be tuned accordingly for the best prediction.

In this paper, the low-fidelity data is generated on the O-mesh in combination with the SA negative turbulence model, since a full 360° could always be computed with this setup. A higher fidelity is achieved with the C-mesh in combination with the k-ω SST model, yet post-stall computations are difficult. By posing the k-ω SST results as the high-fidelity model, a

variable-fidelity model (vfm) can be built, which allows to smoothly transition to SA results, also based on the maximum likelihood prediction. Exemplary in Figure 26 it is seen that at $c_{l,max}$ the results of SA and SST are contradicting each other.



Since high-fidelity data points are still available, the variable-fidelity model still follows this trend. It then gradually drops onto the SA results around 50° angle-of-attack since no high-fidelity data is available anymore. The data interpolated here are $C_l$, $C_d$ and $C_m$ in dependence of the parameter α.

Other ideas for the application may comprise joining CFD and wind tunnel data, which may also be done in a triple-fidelity

setup. Hierarchical Kriging allows to easily chain methods of different fidelities together, since the lower-level model may also be a Hierarchical Kriging model itself.

## 2.6 Framework for the Aeroacoustic Characterization

The aeroacoustic study in this paper is based on a proven and tested tool chain which was primarily presented by Rautmann 2014. These results were further compared to those of other research teams in the BANC-III-Workshop, Category 1,

Trailing-Edge Noise (Herr 2015). The computational results of the above mentioned procedure show a good agreement to numerous measurements of trailing edge noise. The acoustic toolchain is depicted in Figure 4. The employed tools are coloured in blue while the inputs/outputs between the tools are coloured in orange. The final results are highlighted in green.

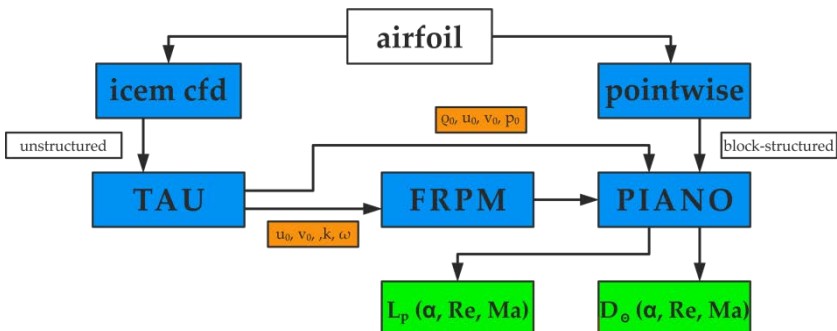

**Figure 4: Framework for the aeroacoustic characterization**

For the aerodynamic part presented one representative Mach and Reynolds number was chosen for the determination of the aerodynamic coefficients. From the aeroacoustic point of view, the flow velocity has a greater impact on the emitted sound than its viscosity. For the aeroacoustic contribution to the aero-data base four different combinations of Mach- and Reynolds numbers are selected, enclosing possible combinations at a full scale wind turbine, see Figure 8. Thus, a new set of RANS computations is necessary.

In a first step, two different 2D meshes around the airfoils are generated. The unstructured C-type mesh is used for a new set of RANS simulations with TAU, while the other one is a block-structured H-type mesh for the computations with the DLR PIANO-code (Perturbation Investigation of Aerodynamic NOise). The simulations with TAU and PIANO run sequentially. The computations of PIANO and the attached FRPM method (Fast Random Particle Mesh), are based on the RANS-output. As the independent variables of PIANO are fluctuating quantities (density $\rho'$, velocities $u'$, $v'$, and pressure $p'$), the

underlying mean-flow quantities $\rho_0, u_0, v_0, p_0$ are required as an input from RANS. The second input for PIANO is provided



by the FRPM tool. Based on the statistics of the kinetic energy of the turbulence $k$, and the specific dissipation rate $\omega$ provided by the k-ω SST turbulence model from (Menter 1994), the FRPM method reconstructs stochastic turbulence (Ewert 2008). The turbulence statistics from the previous RANS solution is realized, again. The FRPM output provides unsteady broadband sound sources on a Cartesian background mesh. These generated fluctuations simultaneously excite the PIANO

code, solving e.g. the Acoustic Perturbation Equations (APE), suppressing the vorticity mode in the solution, (Ewert 2003). The propagated acoustic pressure recorded at 360 microphones located on a circle with a radius of 2.5 $l_c$ around the trailing edge. The microphones collect the acoustic farfield signals which include convective amplification as well as refraction and diffraction effects. This enables the plot of the frequency spectrum $L_p(f)$ for each microphone and the directivity function $D_\Theta$. Using this framework, an aeroacoustic data base is assembled.

**2.7 Aeroacoustic Solver PIANO and FRPM**

The PIANO-Code developed at the Institute of Aerodynamics and Flow Technology is used for Computational AeoroAcoustics (CAA) simulations. As mentioned above, it works in a hybrid two-step procedure. Based on the preceding RANS computations with the TAU code, in the second step time resolved linear propagations equations like the Linearized Euler Equations (LEE) or the above mentioned APE are solved on structured multi-block meshes to compute the sound field.

The CAA solver is based on the 4th order accurate DRP scheme proposed by Tam 1993. The temporal discretization is realized by an alternating two-step low-dissipation, low-dispersion Runge-Kutta (LDDRK) algorithm (Hu 1996). The coefficients are chosen in such a way that the dissipation and dispersion errors are minimized without compromising the stability limits. Combining two alternating steps in the optimization, the dispersion errors are further reduced and a higher order of accuracy is maintained.

The left hand side of the linear propagation equations is stimulated by separately applied sound sources at their right hand side. For the computation of trailing edge noise, a vorticity based sound source is imposed, coupled to the CAA solver. Despite the fact that the APE do not resolve vorticity, they respond on it. The FRPM method realizes these time-resolved fluctuations from time-averaged turbulence statistics. This Gaussian correlated synthetic turbulence represents the fluctuating vorticity according to the turbulence statistics of the RANS solution. The source model is the so called 'Source

A' under application of the Lamb vector.

All acoustic signals can be collected at user-chosen microphones, during simulation. In addition, the temporal and spatial resolved acoustic quantities can be recorded at user-defined Ffowcks-Williams-Hawkings-Surfaces. These signals are to be propagated to the farfield separately in a post processing of the data.

**2.8 Computational Setup for Aeroacoustics**

Based on the procedure presented above, the aeroacoustic investigation was performed according to Figure 4. Meeting as many as possible local flow conditions at different blade elements in the outer 20% of a rotor blade, five different



combinations of Mach and Reynolds numbers were processed. The selected conditions in this paper clamp typical operating conditions of a wind turbine with 20 m Radius. The angles of attack were likewise chosen to cover the expected range during regular operation. Further, two different types of boundary layer are simulated to satisfy different soiling states of the rotor blades. These include a forced transition at 10% along the chord at both sides of the airfoil representing a partially laminar

boundary layer, and a fully turbulent flow along both suction and pressure side. Table 5 gives an overview of the current aeroacoustic data base, building the framework for the presented model, covering the wide range local flow conditions at the blade elements. For the aeroacoustic investigation only airfoils with relative thickness $D/l_c \leq 25\%$ were considered from Table 6.

**Table 5: Variation of angles of attack $\alpha$, boundary layer (BL) type, combinations of Mach and Reynolds number. The Reynolds number is adjusted by variation of chord length ($l_c$).**

| $\alpha$ | Ma | — | Re | BL / transition |
|---|---|---|---|---|
| -4° | 0.15 | — | 751,000 | fully turbulent |
| 0° | 0.32 | — | 751,000 | transition @ 0.1 $l_c$ |
| 4° | 0.15 | — | 2,259,000 | |
| 8° | 0.32 | — | 2,259,000 | |
| 10° | 0.235 | — | 1,500,000 | |

The CFD mesh used for the first step of the hybrid procedure is depicted in Figure 5. In Figure 6 the CAA mesh for the second step is shown. At the trailing edge, the resolution of the small vorticity structures must be realized. The type and resolution of the meshes were set up according to the best practice from Rautmann 2014. The same holds for the choice of the computational parameters used for the DLR codes TAU and PIANO.

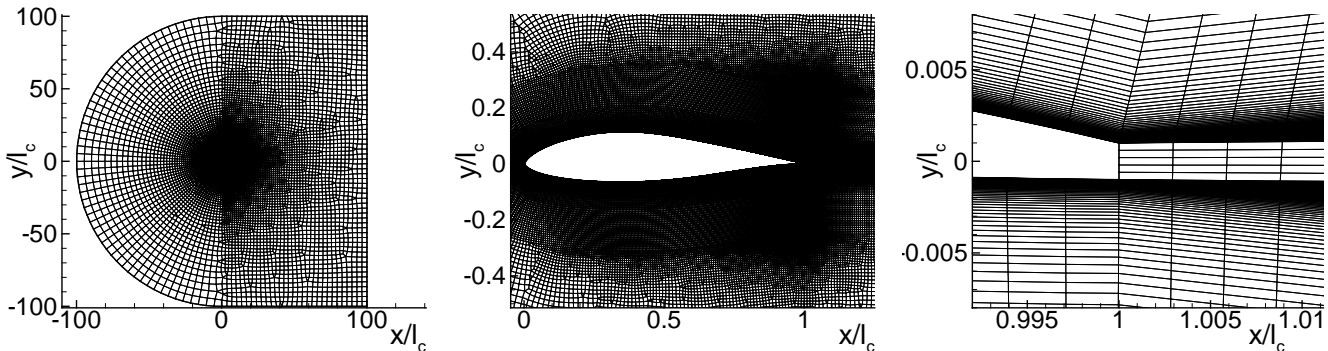

**Figure 5: Unstructured CFD mesh for the aeroacoustic tool chain exemplified for the DU180 consists of 102,500 grid points. The C-topology allows for a resolution of the boundary layer of about 100 prism layers. The mesh extends ±100 $l_c$ in the 2D directions.**



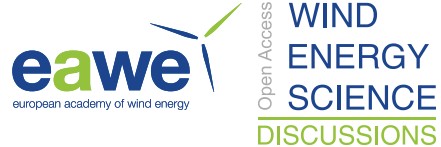

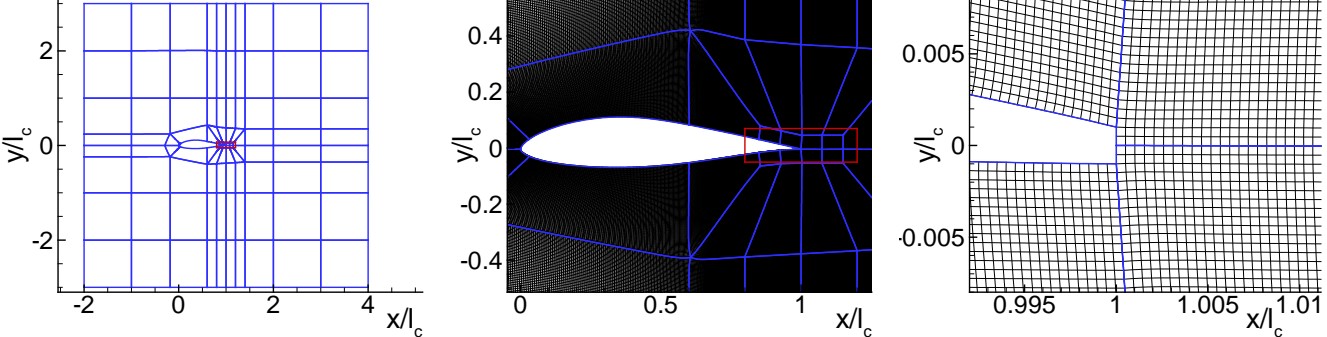

**Figure 6: Block-structured CAA mesh for the aeroacoustic tool chain exemplified for the DU180 consists of 1,200,000 grid points. The number of grid points is kept moderate by combining an H-topology with clamping blocks at the trailing. The mesh is designed to resolve frequencies up to f_max = 10…60 kHz, depending on the actual chord length $l_c$ of the parameter combination of Mach and Reynolds number. The mesh extends ±3 $l_c$ in the 2D directions, centered at the trailing edge.**

### 2.9 Processing of the aeroacoustic data

The resulting—spectrally resolved—sound pressure levels $L_p(f)$ in 2D are normalized to a microphone distance of $r_{norm} = 1$ m (Herr 2012). Further, they are corrected to a span of $s_{norm} = 1$ m to represent standardized 3D spectra (Ewert, 2009).

$$\Delta L_p^{norm} = 10 \ \log_{10}\left(\left(\frac{r}{r_{norm}}\right)\left(\frac{\zeta}{2\pi}\frac{s_{norm}}{r_{norm}}\text{Ma}\right)\right) \tag{1}$$

The actual Mach number (Ma) and the constant ζ=1.4 are taken into account for the transfer to 3D levels. In addition, the frequency spectra are converted to Strouhal spectra.

$$\text{Sr} = f\frac{l_c}{V_{rel}} \tag{2}$$

It is $f$ the narrow band frequency, $l_c$ the chord, and $V_{rel}$ the total inflow velocity. To receive comparable Spectra at different Mach- and Reynolds numbers, the aeroacoustic signals are sampled such that a minimum resolution of $\Delta \text{Sr} = 0.5$ during runtime is guaranteed.

Each such revised spectrum represents the aeroacoustic 3D result at one microphone position for the related airfoil at the associated angle of attack, Mach and Reynolds number, and the selected state of the boundary layer. The objective of the DLR project RoDeO is the design of a new rotor and the estimation of its sound emission. This intention needs for a suitable model of computing the incoming sound at an arbitrary ground position. In a future step, the sound contribution of each blade element—in motion and regarding the directivity function towards the observer—is summed up to a favoured measure of sound emission, see Figure 7 and refer to Rautmann 2017.




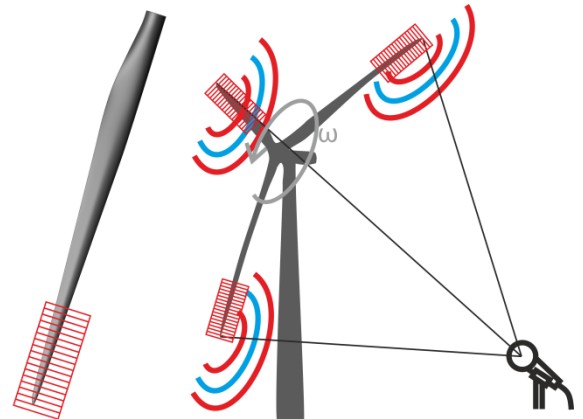

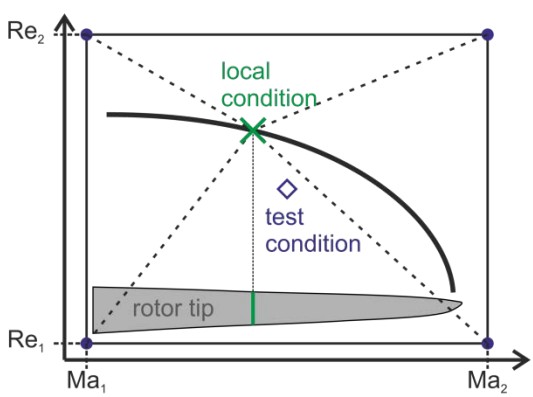

**Figure 7: Partition of the rotor blade into single blade elements and their contribution to the total sound emission of the rotor that is collected at an arbitrary ground position.**

**Figure 8: By bilinear interpolation between four pre-calculated anchor points, local conditions of the rotor can be met. The method is directly verified at a fixed test condition.**

Figure 8 illustrates the idea of bilinear interpolation of the emitted sound from one blade element, based on the four pre-calculated anchor conditions at fixed Mach and Reynolds numbers, as suggested by Faßmann 2017. For each required blade element, the corresponding spectrum and directivity is evaluated. Therefore a scaling model of the sound pressure level $L_p$ is suggested for converting a known spectrum $L_{p,i}$ to an objective spectrum $L_{p,\mathrm{obj}}^{\mathrm{scale}}$. The local Mach and Reynolds numbers are

taken into account. The scaling is based on the scaling of 1/3-octave band spectra used in the BANC II problem statement (Herr 2012). Assuming standardized spectra with constant microphone distance and constant span, the influencing factors are reduced to velocity and viscosity. Thus, the Mach and Reynolds number are selected as Model parameters, skipping the boundary layer thickness $\delta$ known to affect the sound generation at a trailing edge.

$$\Delta L_{p,\mathrm{obj}}^{\mathrm{scale}} = 10 \, \log_{10}\left( \left(\frac{\mathrm{Ma_{obj}}}{\mathrm{Ma}_i}\right)^n \left(\frac{\mathrm{Re_{obj}}}{\mathrm{Re}_i}\right)^m \right) \tag{3}$$

The model exponents $n$ and $m$ may be estimated or determined separately by regression of the results at same Reynolds (for $n$) and same Mach number (for $m$). The exponents tend to $2 < n < 4$ and $0 < m < 2$. One should expect a typical scaling of $n + m + 1 = 5$ for the overall sound pressure level (OASPL) and $n + m = 4$ for the narrowband spectra. The results shown in Section 3.3 Aeroacoustic Results) will be computed for $n = m = 2$ without individual determination of the exponents to emphasize the general capability of the suggested method. If necessary, the levels might further be shifted according to the

ambient temperature and density. With intent to compare different rotors at the same ambient conditions, this final offset is omitted.

Based on the standardized Spectra $L_p^{\mathrm{norm}}(\mathrm{Sr})$, according to eqns (eq.1) and (eq.2), the levels at the anchor conditions $(\mathrm{Ma}_i, \mathrm{Re}_i)$ are shifted to the objective conditions $(\mathrm{Ma_{obj}}, \mathrm{Re_{obj}})$, following eqn (eq.3). All resulting $L_{p,i,obj}^{\mathrm{norm,\,scale}}$ are bilinearly interpolated—in the guise of Power Spectral Densitiy (PSD)—to emulate the acoustic result at the desired conditions

$(\mathrm{Ma_{obj}}, \mathrm{Re_{obj}})$ in terms of PSD or $L_{p,\mathrm{obj}}^{\mathrm{emul}}(\mathrm{Sr})$.





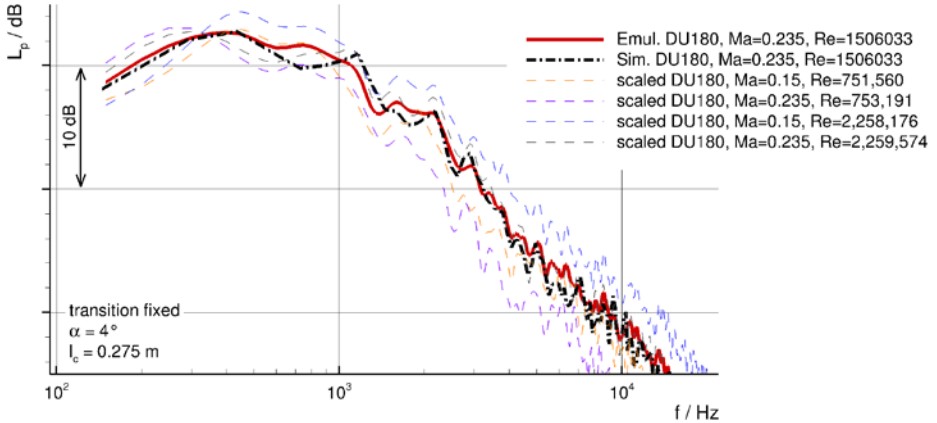

**Figure 9: Emulated and simulated frequency spectrum for the DU180 airfoil for the above marked reference test condition at 4° angle of attack with fixed transition BL. The exponents were chosen to n=3 and m=1. The dashed lines depict the appropriately rescaled spectra at the anchor points according to eq.3.**

In Figure 9 the comparison between the simulated frequency spectrum $L_{p,\mathrm{obj}}^{\mathrm{sim}}(f)$ at the test condition and the corresponding emulated spectrum $L_{p,\mathrm{obj}}^{\mathrm{emul}}$ is shown. Over a wide frequency range the emulation technique provides good agreement between simulated and emulated results with accuracy of about 2 dB for the individual computed values of $n = 3$ and $m = 1$.

## 3 Aerodynamic & Aeroacoustic Results

**Table 6: Investigated airfoils with given relative thickness ($D/l_c$), original and adapted relative trailing edge thickness ($d/l_c$), and the given identifier of the adapted airfoil.**

| airfoil | owner | D/l$_c$ | original d/l$_c$ | adapted d/l$_c$ | identifier |
|---|---|---|---|---|---|
| DU-96-W-180 | DTU, Denmark | 18.0% | 0.24% | 0.20% | DU180 |
| DU-93-W-210 | DTU, Denmark | 21.0% | 0.38% | 0.30% | DU210 |
| DU-91-W2-250 | DTU, Denmark | 25.0% | 0.42% | 0.40% | DU250 |
| DU-97-W-300 | DTU, Denmark | 30.0% | 0.50% | 1.00% | DU300 |
| DU-99-W-350 | DTU, Denmark | 35.0% | 0.58% | 3.50% | DU350 |
| FFA-W3-211 | FFA, Sweden | 21.1% | 0.26% | 0.30% | FFA211 |
| FFA-W3-241 | FFA, Sweden | 24.1% | 0.75% | 0.40% | FFA241 |
| FFA-W3-301 | FFA, Sweden | 30.1% | 1.83% | 1.50% | FFA301 |
| FFA-W3-360 | FFA, Sweden | 36.0% | 2.90% | 3.50% | FFA360 |
| LN118 | CQU China, DTU Denmark | 18.0% | 0.20% | 0.20% | LN118 |





One important goal of this paper is the comparison of aeracoustic characteristics among different airfoils of equivalent thickness. Because the relative trailing edge thickness ($d/l_c$) is one of the key parameters for trailing edge noise, the original airfoil geometries had to be slightly adapted in order for airfoils of the same thickness to be directly comparable. The simulated airfoils and its modifications are displayed in Table 6. In the case of validation the original geometry has been

used. Thus the complete airfoil identifier (e.g. DU-97-W-300) has been employed for captions and in the text. On the other hand the short identifier always indicates that for the underlying investigation the modified airfoil has been applied.

### 3.1 Validation of Aerodynamic Approach

Since the computation of airfoil polars in the post-stall region is a non-trivial task, it is mandatory to validate the proposed method before starting the computations for the airfoil catalogue. That means that the available methods have to be

compared exemplarily to experimental data. In this case the various available methods will be compared to the experimental data of the well-known DU-93-W-210 and the DU-97-W-300 airfoil. W.A. Timmer from the Technical University of Delft kindly provided the data for this purpose. Figure 10, Figure 14 and Figure 15 depict the lift, drag and moment coefficient over the angle of attack of the DU-93-W-210. Besides the experimental data, three CFD computations with different numerical settings and data obtained from an XFOIL computation are shown. Below and throughout the paper the three

different methods will be termed as: Method 1 (C-topology mesh + k-ω SST or only SST), Method 2 (O-topology mesh + k-ω SST) and Method 3 (O-topology mesh + SA negative or only SA).

Lift comparison of DU-93-W-210 between -30° and 30° angle of attack (Figure 10):

It can be seen that all methods perform very well in the linear range which extends between -8° and 8° degrees angle of attack. Major differences occur in the positive and negative stall region. The XFOIL results show hardly any stall behaviour.

The two CFD computations on the O-topology mesh overpredict the $C_{lmax}$ value by 0.3 to 0.5 i.e. 20% and 35% respectively. Only the CFD solution using Method 1 compares very well for $C_{lmax}$ with the experimental value and compares also fairly well in the post-stall region. The pressure distributions over the airfoil chord are compared in Figure 11. While $c_{P,min}$ is predicted similarly for Method 1 and 3, the prevision of the suction peak is much higher for Method 2. Subsequently the results in the region in which the pressure recovery takes place (20% to 40% airfoil chord) differ in that Method 1 predicts a

stronger pressure recovery with the consequence of reaching a smaller underpressure at 40% chord. Even higher differences can be observed on the aft part of the airfoil. The very different pressure distributions are due to a very different prediction of the trailing edge vortex which has grown substantially in comparison to lower angles of attack and now covers more than 50% of the airfoil chord. Having a closer look at the convergence and the resulting lift coefficients in form of a time series in Figure 12 reveals that Method 2 and Method 3 have not converged to a satisfactory level resulting in strong oscillations of

the lift coefficient. This behaviour has been observed to be similar for a vast number of profiles within a certain range of angles of attack. Therefore the amplitude of the lift, drag and moment coefficient has always been computed during the post-processing and is plotted as shown in Figure 13. The visualization of the amplitudes has thereby been used as a trust indicator.



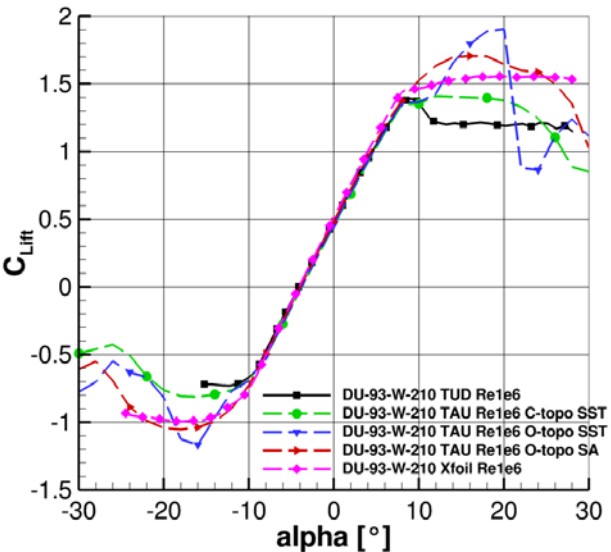

**Figure 10: Comparison of lift polar for DU-93-W-210**

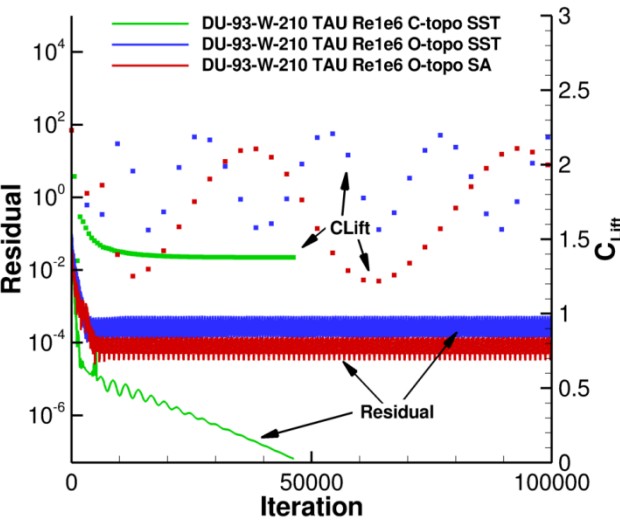

**Figure 12: Residual and lift coefficient for alpha = 20°**

**Figure 11: Geometry and pressure distribution for alpha = 20°**

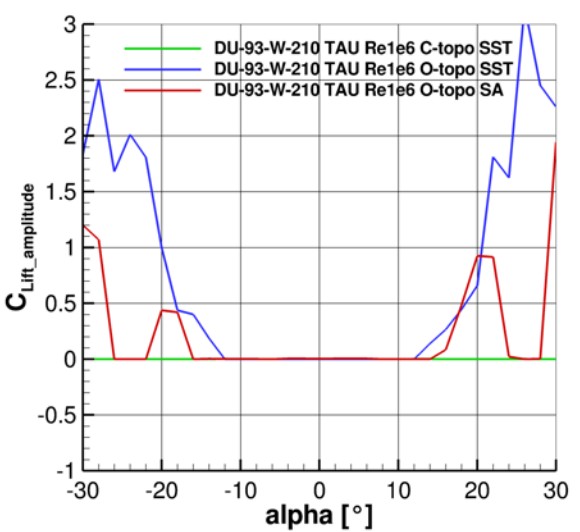

**Figure 13: Comparison of lift amplitudes for DU-93-W-210**

Small oscillations have been considered trustworthy and have been treated by an averaging process (see Section 2.4), while bigger oscillations have usually been eliminated from the dataset or been smoothed by the successive interpolation process (see Section 2.5). For the $C_{lmin}$ value and the post-stall region for negative angles of attack similar conclusions for the comparison between the CFD results and the experimental data have been found. In this region Method 1 also performs best

5    in comparison with the experimental data.

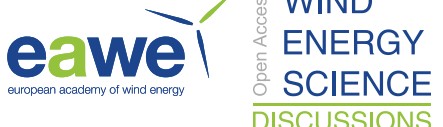



Drag comparison of DU-93-W-210 between -30° and 30° angle of attack (Figure 14):

As could already be seen for the lift polar the various methods also agree very well for the drag values in the range of -8° to 8° angle of attack. For angles of attack greater than 8° or smaller than -8° the drag rise is predicted very different among the various methods. The least accurate is XFOIL because it under predicts the experimental value the most. Method 2 is found

on the other end because it also over predicts the drag coefficient by a great amount. Method 1 and Method 3 perform best at higher angles of attack. Unfortunately no measurements exist for angles of attack smaller than-16°. The graph for the drag amplitude has been omitted since the conclusions are equivalent to the ones in the lift case.

Moment comparison of DU-93-W-210 between -30° and 30° angle of attack (Figure 15):

For the moment coefficient the accordance among the various methods between -8° and 8° is less congruent than for the lift

and drag coefficient. While XFOIL tends to slightly under predict the experiment, results achieved with Method 1 slightly lie above. The other methods predict values in between and are therefore closer to the experiment. The differences are relatively small overall. For smaller angles of attack XFOIL, Method 1 and Method 3 perform almost equally. For higher angles of attack Method 1 proofs once more to perform best. Method 3 only slightly deviates from the other methods and experiment.

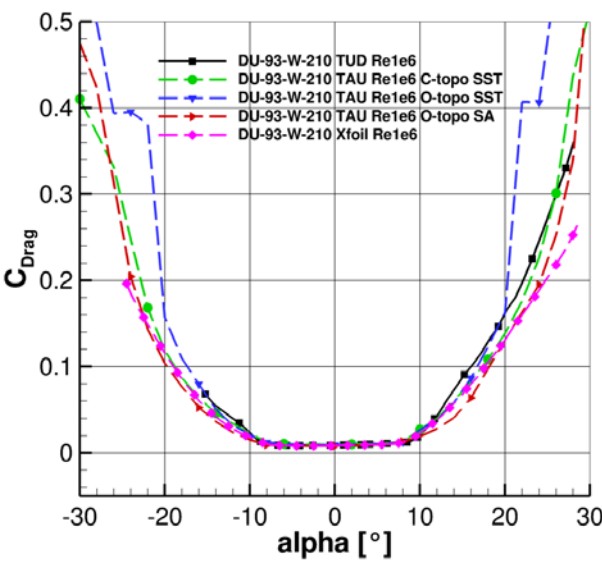

**Figure 14: Comparison of drag polar for DU-93-W-210**

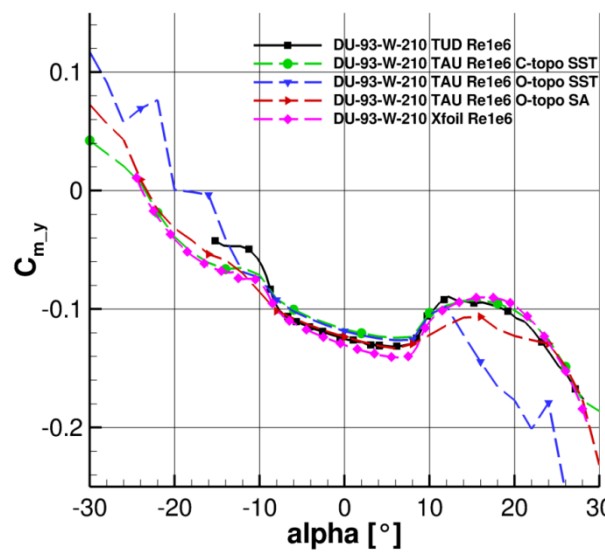

**Figure 15: Comparison of moment polar for DU-93-W-210**

Figure 16 to Figure 25 deal with the comparison of the computational methods and the experiment for the 30% thick airfoil

DU97-W-300. While Figure 16 to Figure 21 focus on the range of angles of attack between -30° and 30°, Figure 22 to Figure 25 reveal the differences between the various methods for the complete range of angles of attack.

Lift comparison of DU-97-W-300 between -30° and 30° angle of attack (Figure 16):

The geometry of the DU-97-W-300 differs quite severely from the DU-93-W-210. One of its most prominent features is the thick lower surface with its highest thickness around 28% chord length. Naturally the linear range of this airfoil is less

extensive than for the DU-93-W-210. In concordance with all methods this range can be defined between -4° and 8° angle of



attack. Within this range all methods agree fairly well. As in the case of the previous airfoil most methods over predict the $C_{lmax}$ value by 15% to 30%. Furthermore Method 2 and Method 3 compute $C_{lmax}$ to be reached for an angle of attack around 16°. Method 1 and XFOIL localize the maximum lift coefficient around 14° which is in better accordance with the experiment.

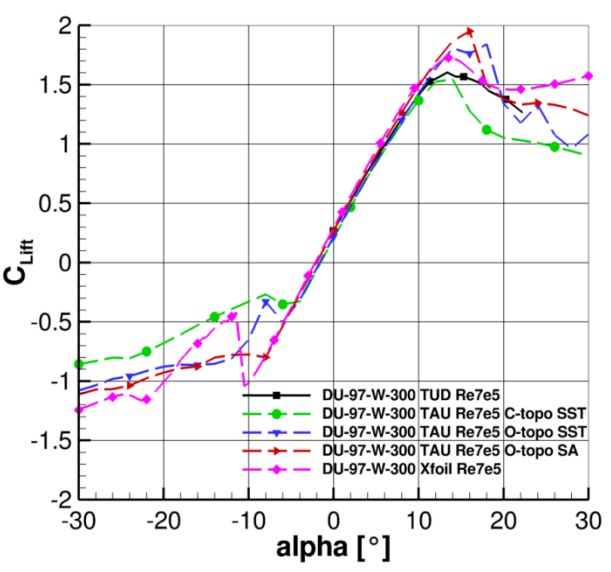

**Figure 16: Comparison of lift polar for DU-97-W-300**

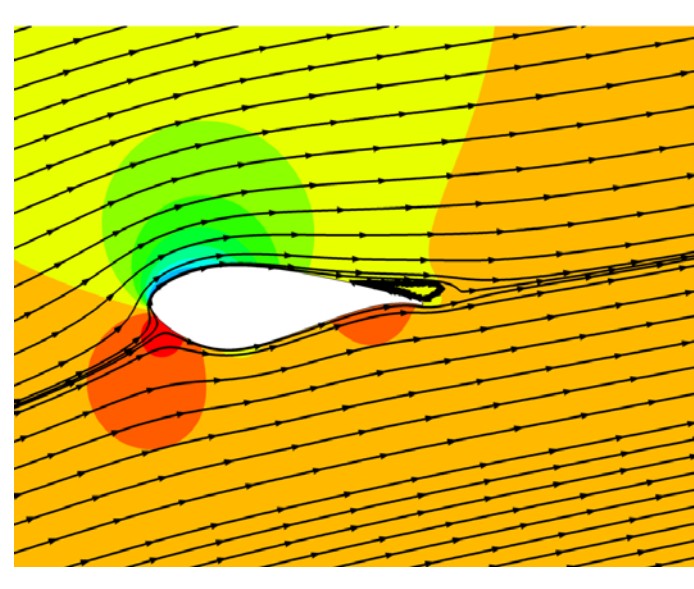

**Figure 17: Pressure distribution and streamlines around DU-97-W-300 with k-ω SST + C-topology mesh for α = 14°**

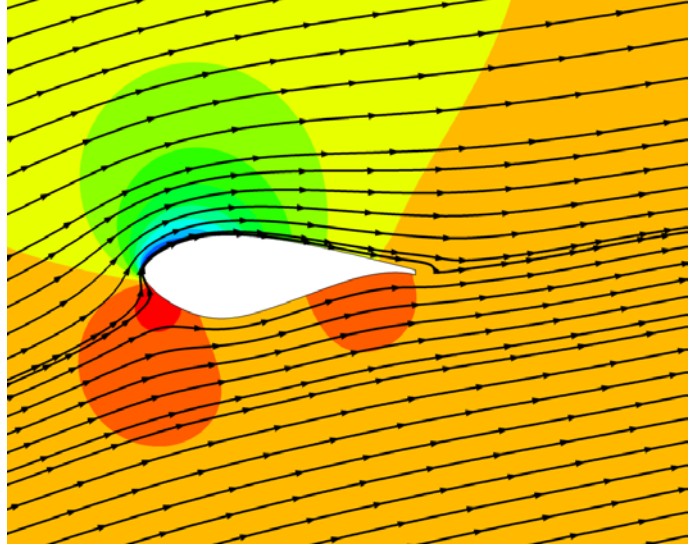

**Figure 18: Pressure distribution and streamlines around DU-97-300 with SA neg + O-topology mesh for α = 14°**

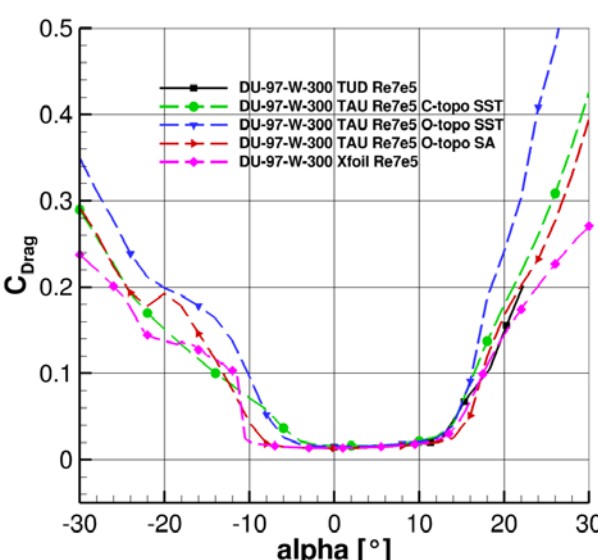

**Figure 19: Comparison of drag polar for DU-97-W-300**





Figure 17 shows the pressure distribution around the airfoil computed with Method 1 for an angle of attack of 14°. The equivalent plot for Method 3 is provided in Figure 18. While in the case of the k-ω SST model an incipient trailing edge separation can be seen, the flow is still fully attached in the case of the SA negative model. This is due to the high suction peak at the nose predicted by the SA negative model. The lower pressure at the nose leads to a stronger acceleration of the

flow on the first 30% of the suction side and consequently to a higher stream wise velocity on the complete upper surface. Therefore the boundary layer stays attached in case of the SA negative model, while it separates shortly before the trailing edge in case of the k-ω SST model. Hence the $C_l$ for an angle of attack of 14° is predicted about 0.3 higher by Method 3 in comparison with Method 1. In fact the over prediction of the $C_{lmax}$ by the SA model in regions with adverse pressure gradients is quite known and various researchers (e.g. Medida 2013) have tried to improve the model for those kinds of

situations. The differences persist for angles of attack greater than $\alpha_{max}$. In this case the flow is separated for both models with k-ω SST always predicting a bigger separation area due to the fact that the kinetic energy in the boundary layer as predicted by the k-ω SST is smaller than with the SA negative model and therefore separates earlier. The difference in the $C_l$ value stays about the same as for $C_{lmax}$, but care has to be taken since the convergence of the SA negative model is not optimal above $C_{lmax}$ as can be seen by the drag amplitude shown in Figure 20. Although for an angle of attack of e.g. $\alpha = 20°$

the lift coefficient computed with Method 3 matches best the experimental measurements, it must be noted that the measurements are based only on the pressure sensors and that interference effects with the boundary layer from the wind tunnel wall might affect the measurements to a greater extent than for smaller angles of attack. Unfortunately the exact procedure for obtaining the experimental values is unknown.

For the region of negative angles of attack also huge differences can be observed. While separation with the SA negative

model begins at an angle of attack of $\alpha = -8°$, the separation in case of the k-ω SST model starts already for $\alpha = -4°$. Unfortunately no experimental data exist in this region of the lift curve. For a Reynolds number of 1e6 (not shown here) the experimental values match best the results obtained by Method 3. But the transition in the experiment has been influenced by vortex generators. Therefore an exact comparison is difficult to accomplish.

Drag comparison of DU-97-W-300 between -30° and 30° angle of attack (Figure 19):

As in the case of the previous airfoil the methods agree very well in the linear range. For angles of attack greater than 8° Method 1 and Method 3 perform almost equally well. Differences in $C_d$ are only minor, although the drag amplitude depicted in Figure 20 indicates that the convergence of the computations between 14° and 24° angle of attack is not optimal. As stated before the amplitude is only an indicator of potential convergence difficulties. Apparently in this case the oscillations are small and the averaging process seems to be valid.

On the contrary deviations between Method 1 and Method 3 become bigger for negative angles of attack. Especially around -20° angle of attack the drag polar computed with Method 3 features a discontinuous behavior that looks suspicious. An examination of the drag amplitude in this region (Figure 20) indicates stronger oscillations and therefore a poorer convergence of each calculation. Although no experimental data is available in this region, the results obtained with Method 1 seem to be more trustworthy.



Method 2 and XFOIL show less agreement with the experimental data for both negative and positive angles of attack outside the linear range.

Moment comparison of DU-97-W-300 between -30° and 30° angle of attack (Figure 21):

The variance of the results among the various methods for the moment coefficient is greater as for the other two coefficients

and also in comparison with the DU-93-W-210 airfoil. Even in the linear range larger differences are visible. XFOIL and Method 3 tend to under predict the experimental value. Method 2 and Method 3 show good agreement between 0° and 14°. The subsequent drop of $C_m$ in the experimental data set looks somewhat suspicious, since it occurs very sudden. Moreover for α = 20° three out of four methods coincide with the experimental value. And the polar curve predicted by XFOIL or Method 1 look somewhat smoother between 14° and 20° angle of attack. The drag amplitude in Figure 20 (which is similar

to the moment amplitude) suggests that the computations of Method 2 outside the linear range did not converge very well. Thus it is not surprising that the polar curve for Method 2 looks very uneven in these regions.

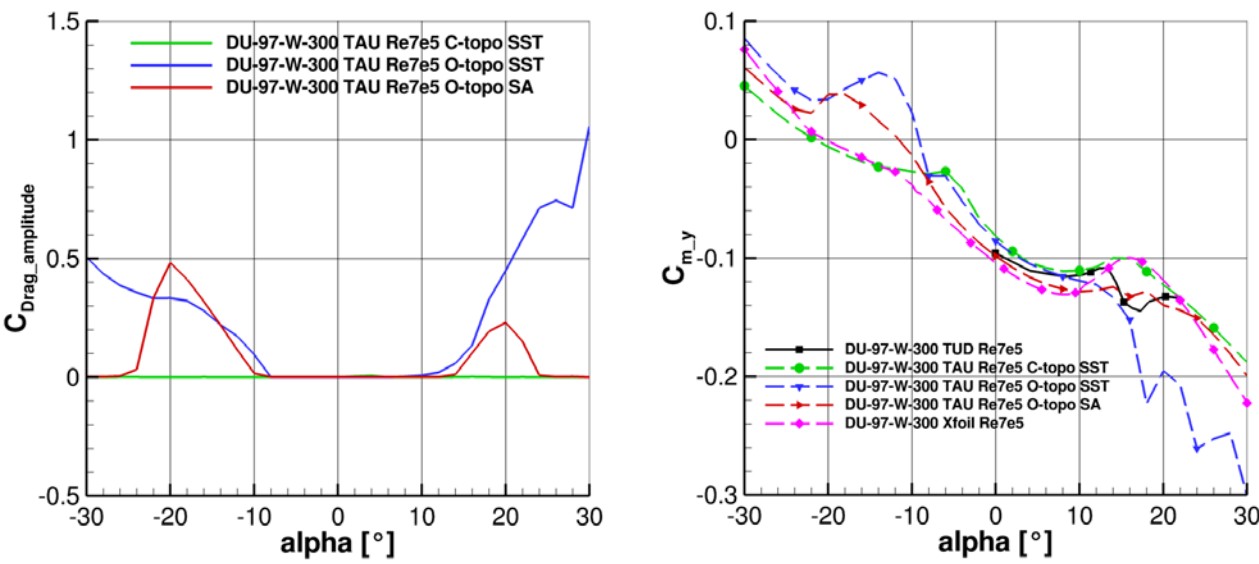

**Figure 20: Comparison of drag amplitudes for DU-97-W-300**   **Figure 21: Comparison of moment polar for DU-97-W-300**

Figure 22 to Figure 25 focus on the discussion of the aerodynamic coefficients of the DU-97-W-300 **outside** the region of -30° and 30°.

Lift comparison of DU-97-W-300 **outside** of -30° and 30° angle of attack (Figure 22):

At first glance the results from the different simulations agree relatively well for a wide range of angles of attack. Eight different regions can be detected along the complete range of angles of attack. Region 1: α = -180° to -160°, Region 2: α = -160° to -100°, Region 3: α = -100° to -70°, Region 4: α = -70° to -30°, Region 5: α = 30° to 70°, Region 6: α = 70° to 100°, Region 7: α = 100° to 150° and Region 8: α = 150° to 180°. In Region 3 and Region 6 the agreement among all methods is very good and the results compare very well with the experimental data. The accordance among the methods is somewhat

surprising because only Method 3 appears to be well converged. Figure 24 (plot of lift amplitude looks similar) reveals that

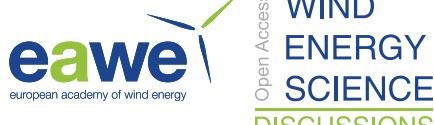

Method 1 and Method 2 suffer from severe convergence problems in Region 3 and Region 6. Despite this fact the agreement is very good. In Region 1 and Region 8 the discrepancies between the methods become greater, but the agreement with the experimental data is still fair. Especially the results from Method 3 are convincing because the peaks in the experimental data at various angles of attack (i.e. -164°, -160°, 155°, 165° and 175°) are fairly well captured.

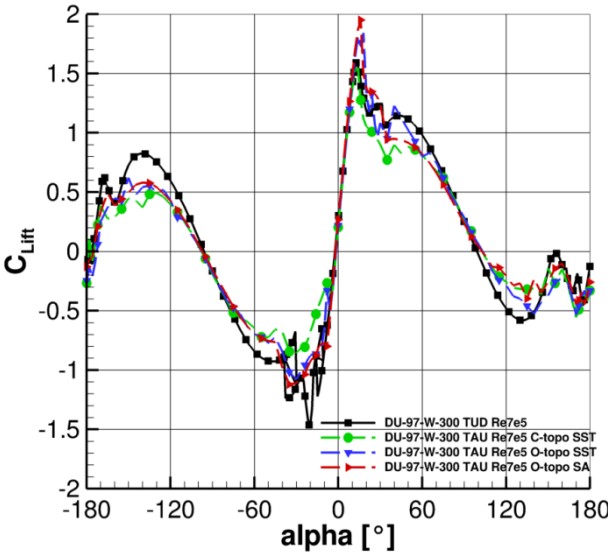

**Figure 22: 360° lift polar for DU-97-W-300**

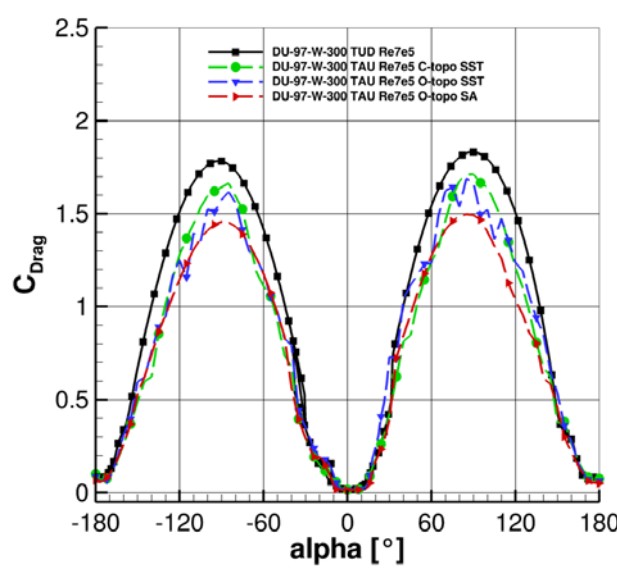

**Figure 23: 360° drag polar for DU-97-W-300**

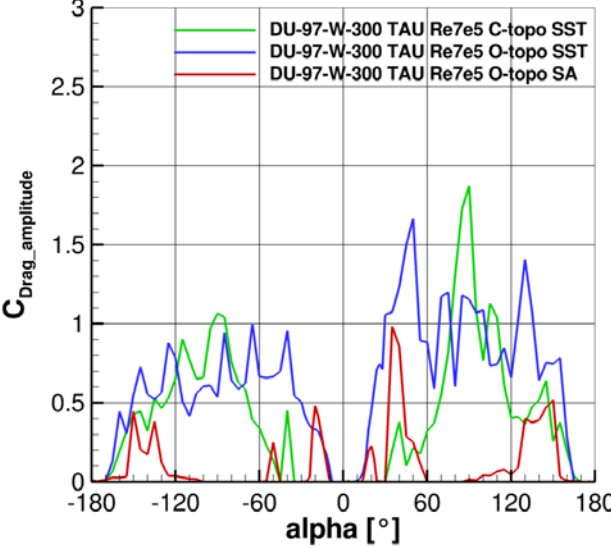

**Figure 24: Comparison of drag amplitudes for DU-97-W-300**

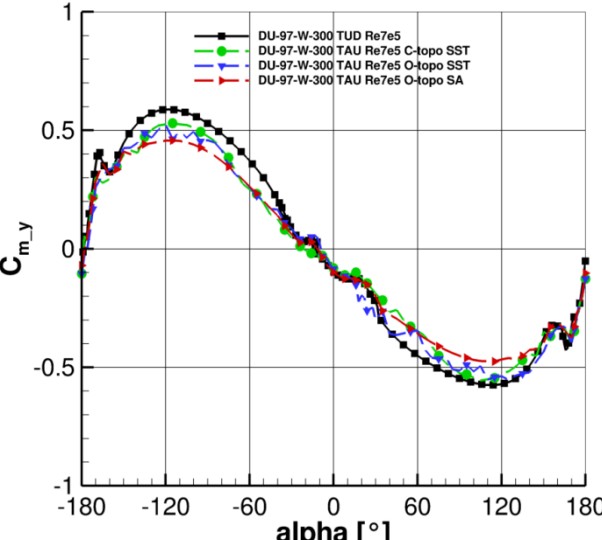

**Figure 25: 360° moment polar for DU-97-W-300**





Less satisfactory are the results from Region 2, Region 4, Region 5 and Region 7. Indeed in these regions the methods represent the trend of the experimental data fairly well, but they are incapable of matching the absolute value. In average the predicted value is about 0.3 smaller in comparison to the measured value.

Drag comparison of DU-97-W-300 **outside** of -30° and 30° angle of attack (Figure 23):

The results with respect to the drag polar are also a bit peculiar. From an examination of Figure 23 Method 1 clearly performs best because the method captures the maximal drag value best. Method 3 stays more than 0.2 below the value predicted by Method 1. Nevertheless this might also be coincidence because as can be seen from Figure 24 the convergence of Method 1 outside -30° to 30° is quite unsatisfactory. The second reason why Method 3 is preferred in this area is that the drag curve is in general smoother than the drag curve computed with Method 1 and especially in comparison with

Method 2.At this point the reader should note that although all methods are not capable of predicting the exact absolute value, all methods predict a smaller drag value for α = -90° than for α = 90°. A feature due to the curvature of the airfoil and that none of the approximation methods analyzed by Skrzypiński 2014 is able to predict.

Moment comparison of DU-97-W-300 **outside** of -30° and 30° angle of attack (Figure 25):

Similar as in the case of the drag polar Method 1 seems to obtain the best results for the moment polar as well. On the other

hand Method 3 manages to better capture the kink around -160° and 160°. Nevertheless Method 3 is chosen for further investigations in this area because of its better convergence and in general better smoothness.

As can be seen neither computational approach was superior to all others for all angles of attack. Therefore both computational setups are used – each setup is applied in the region where it performs best. Afterwards the datasets are fused

to one complete dataset as proposed in Section 2.5 (Interpolation Model for Data Fusion (POT)).

### 3.2 Aerodynamic Polars

**DU180:**

Re = 1e6, Ma = 0.12

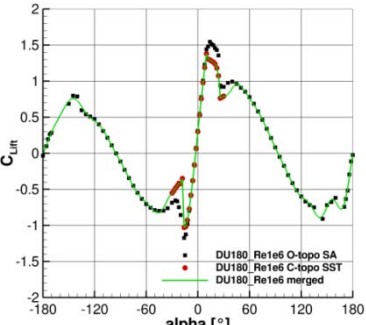 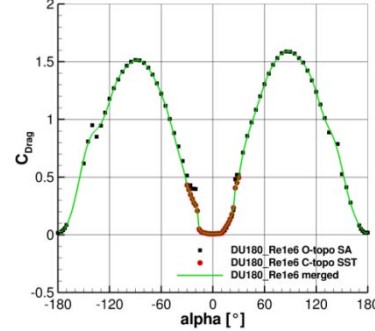 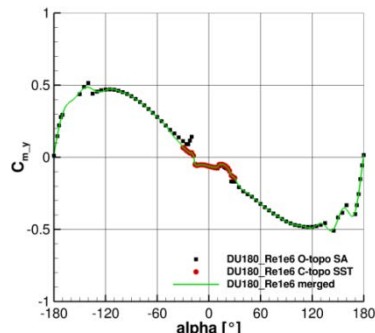

**Figure 26: Lift polar of DU180**    **Figure 27: Drag polar of DU180**    **Figure 28: Moment polar of DU180**




**DU210:**

Re = 1e6, Ma = 0.12

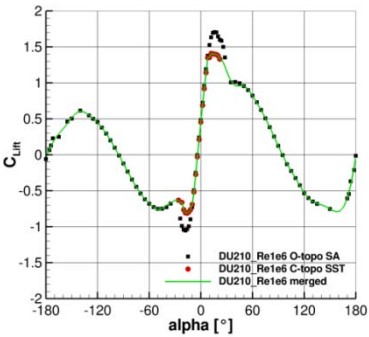
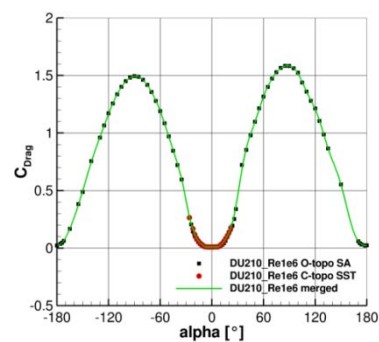
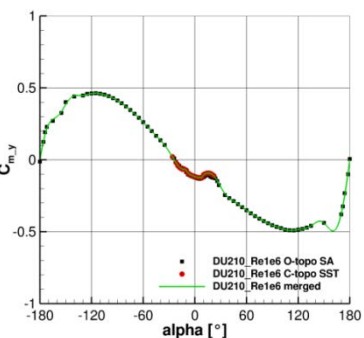

**Figure 29: Lift polar of DU210**    **Figure 30: Drag polar of DU210**    **Figure 31: Moment polar of DU210**

**DU250:**

Re = 1e6, Ma = 0.12

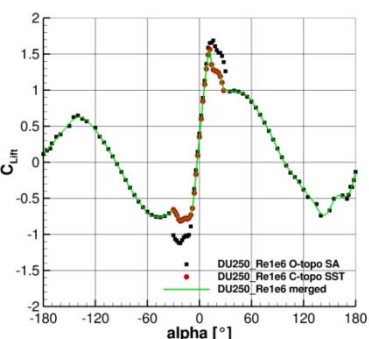
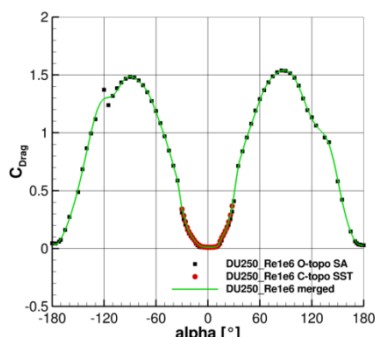
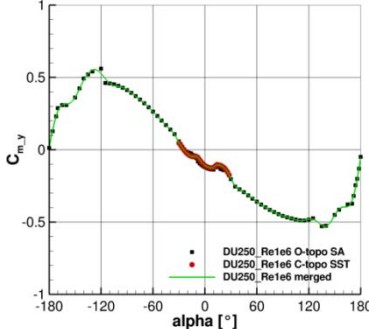

**Figure 32: Lift polar of DU250**    **Figure 33: Drag polar of DU250**    **Figure 34: Moment polar of DU250**

5    **DU300:**

Re = 1e6, Ma = 0.12

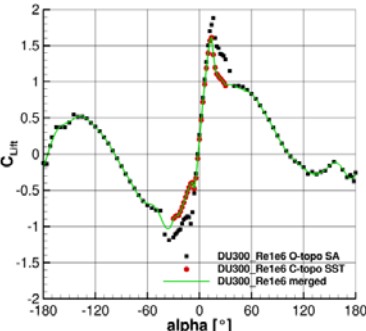
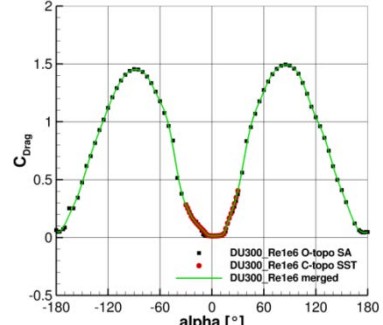
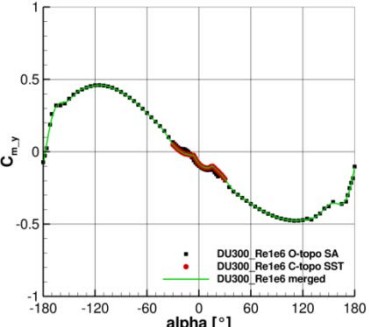

**Figure 35: Lift polar of DU300**    **Figure 36: Drag polar of DU300**    **Figure 37: Moment polar of DU300**





**DU350:**

Re = 1e6, Ma = 0.12

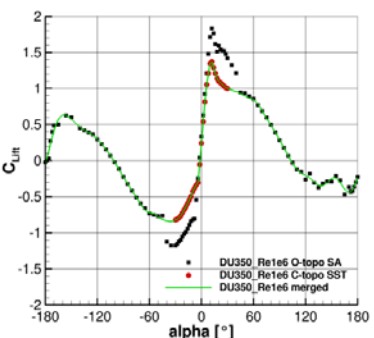

**Figure 38: Lift polar of DU350**

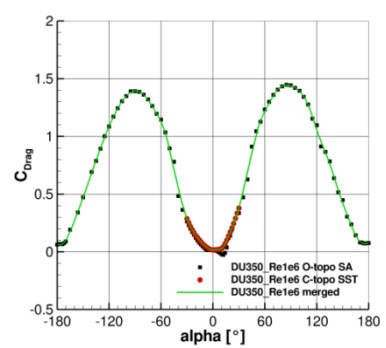

**Figure 39: Drag polar of DU350**

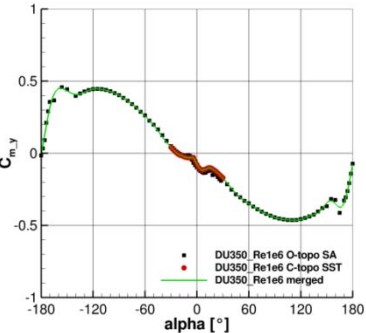

**Figure 40: Moment polar of DU350**

**FFA211:**

Re = 1e6, Ma = 0.12

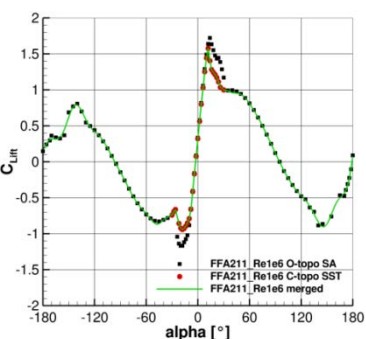

**Figure 41: Lift polar of FFA211**

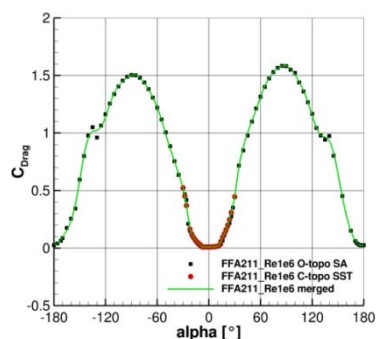

**Figure 42: Drag polar of FFA211**

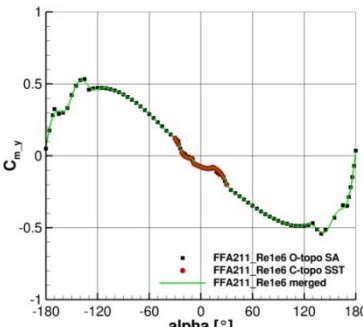

**Figure 43: Moment polar of FFA211**

5  **FFA241:**

Re = 1e6, Ma = 0.12

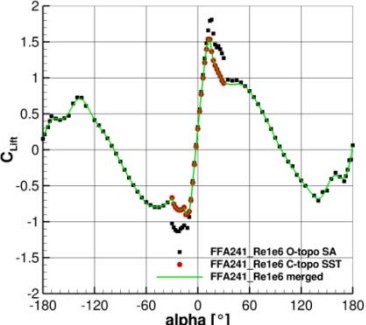

**Figure 44: Lift polar of FFA241**

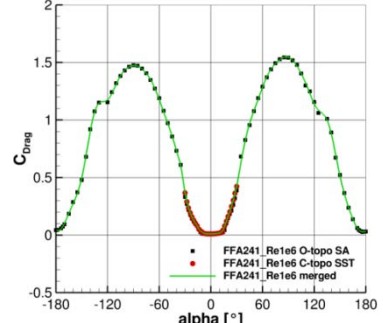

**Figure 45: Drag polar of FFA241**

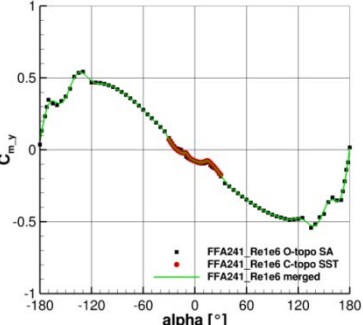

**Figure 46: Moment polar of FFA241**




**FFA301:**

Re = 1e6, Ma = 0.12

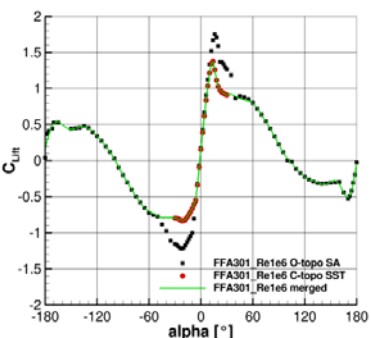
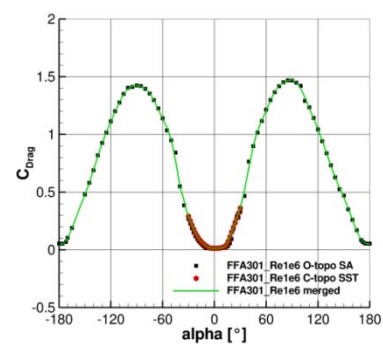
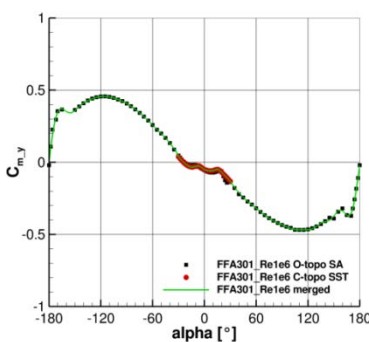

**Figure 47: Lift polar of FFA301**  **Figure 48: Drag polar of FFA301**  **Figure 49: Moment polar of FFA301**

**FFA360:**

Re = 1e6, Ma = 0.12

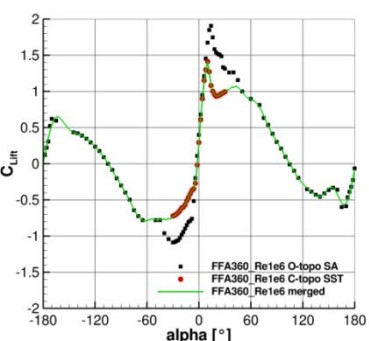
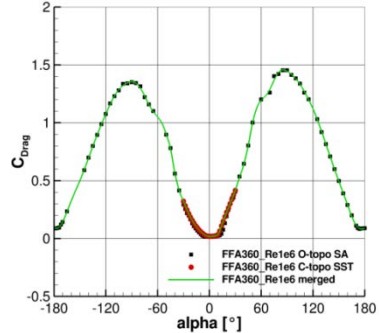
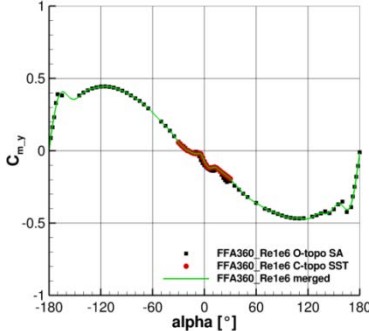

**Figure 50: Lift polar of FFA360**  **Figure 51: Drag polar of FFA360**  **Figure 52: Moment polar of FFA360**

5 **LN118:**

Re = 1e6, Ma = 0.12

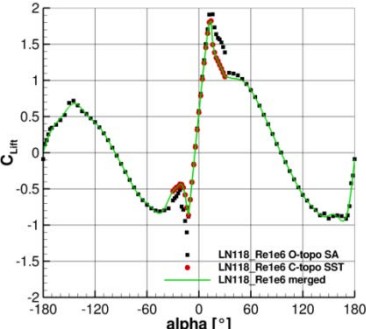
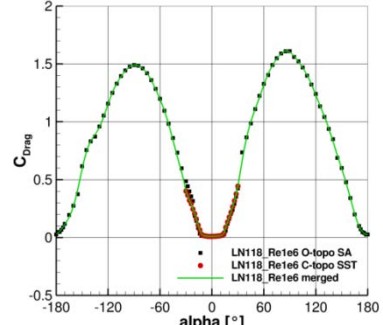
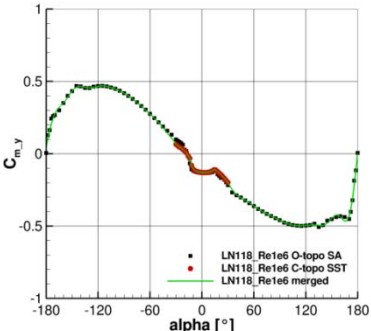

**Figure 53: Lift polar of LN118**  **Figure 54: Drag polar of LN118**  **Figure 55: Moment polar of LN118**



## 3.3 Aeroacoustic Results

During computation the trailing edge sound of each case specified in Table 5 is recorded by 360 microphones. These data were post processed and the narrow band spectra of the microphone perpendicular below the inclined trailing edge are evaluated in detail, see Figure 56. The emulation process characterized in Section 2.9 Processing of the aeroacoustic data) is

applied to these data. The associated results are presented in Section 3.3.1 Spectral results).

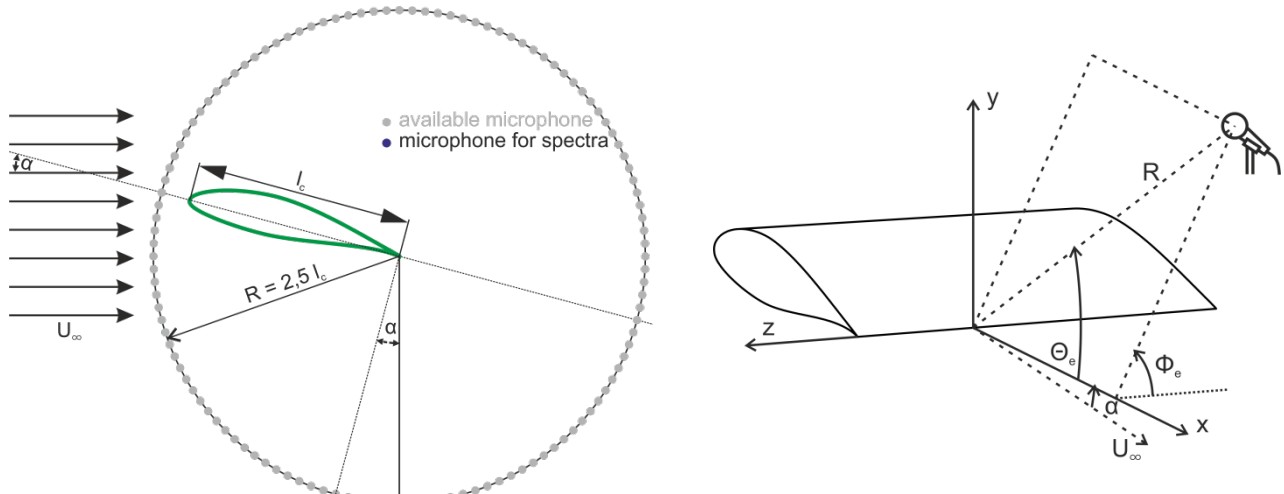

**Figure 56: Schematic of all available microphones and the selected microphone for the evaluation of the spectra perpendicular below the trailing edge at all angles of attack.**

**Figure 57: Definition of the directivity functions according to Brooks 1989.**

Further, the spectral simulation results are summed up to an overall sound pressure level for a selected frequency range. These results are shown in Section 3.3.2 Overall sound pressure levels). They emphasize the dependency of the noise from the airfoil shape and the angle of attack.

Evaluation of all microphones located circular to the trailing edge allows for the plot of the directivity function $D_\Theta$. This

function is part of the total radiation characteristic of a rotor. With regard to Schlinker 1981, Brooks 2001 and Oerlemans 2009, this characteristic can be expressed as

$$D = D_\Theta \, D_\Phi \, \omega/\omega_o \, . \qquad (4)$$

The two spatial directivity functions $D_\Theta$ (from the simulation) and $D_\Phi = \sin^2 \Phi$ are illustrated in Figure 57. The last part of $D$ considers the Doppler related frequency shift. This Doppler factor $\omega/\omega_o$ has to take regard to the relative motion between

source, observer and flow in the case of the final emission of the wind turbine, see Faßmann 2017. This factor is not yet specified, as the CAA already includes the relative motion between airfoil and the flow. However, Section 3.3.3 Directivity functions) will display some of the simulated directivity functions $D_\Theta$.




### 3.3.1 Spectral results

In Figure 58 to Figure 63 the emulated spectra by the bilinear interpolation are compared to the simulated spectra at the same test conditions. The model exponents $n$ and $m$ are chosen to $n = m = 2$. The regression of these exponents is still open. Nonetheless, the results show a good agreement with accuracy of about 3 dB over a large range of frequencies and angles of

5  attack.

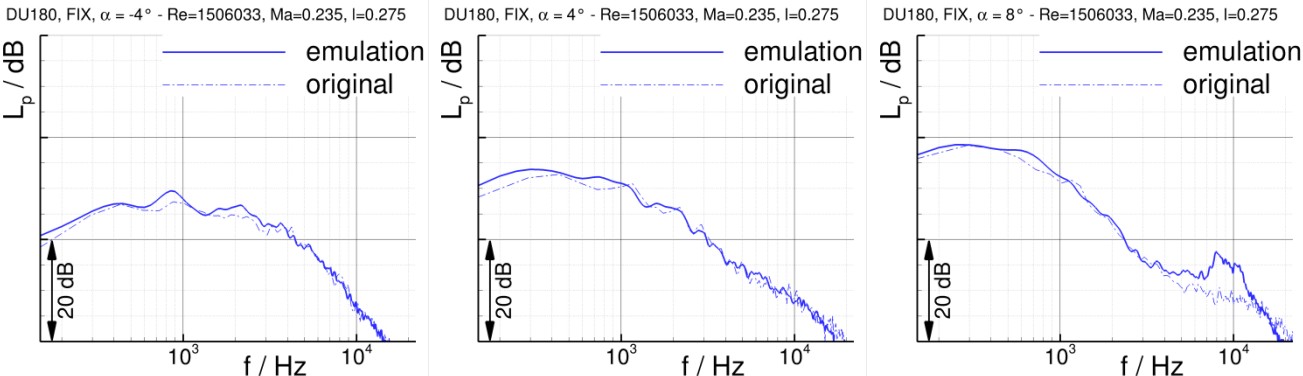

**Figure 58: Comparison of emulated spectra according to eq.3 with $n = m = 2$ and originally simulated spectra at the test condition with fixed transition (FIX) for the airfoil DU180 at $\alpha = -4°, 4°, 8°$.**

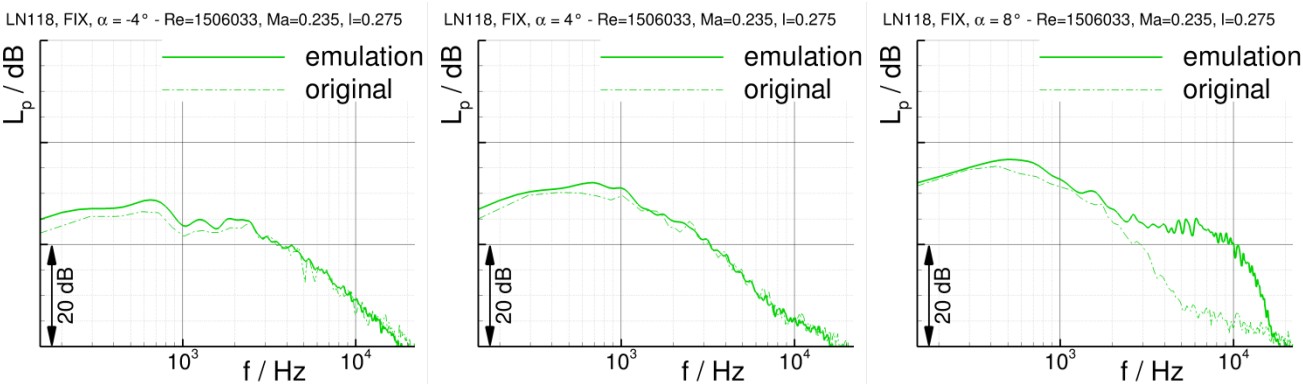

**Figure 59: Comparison of emulated spectra according to eq.3 with $n = m = 2$ and originally simulated spectra at the test condition with fixed transition (FIX) for the airfoil LN118 at $\alpha = -4°, 4°, 8°$.**

If the thinner airfoils are inclined with 8°, deviations show up between emulated and simulated spectra at frequencies above 5 kHz. This can be ascribed to the contribution of single computations at the anchor condition at high Mach and low Reynolds number and with a chord of $l_c = 0.101$m at the same time, see Table 5. The RANS solution in these cases does

10  not show any abnormalities, the flow is fully attached, but the contribution of high frequencies is dominant.




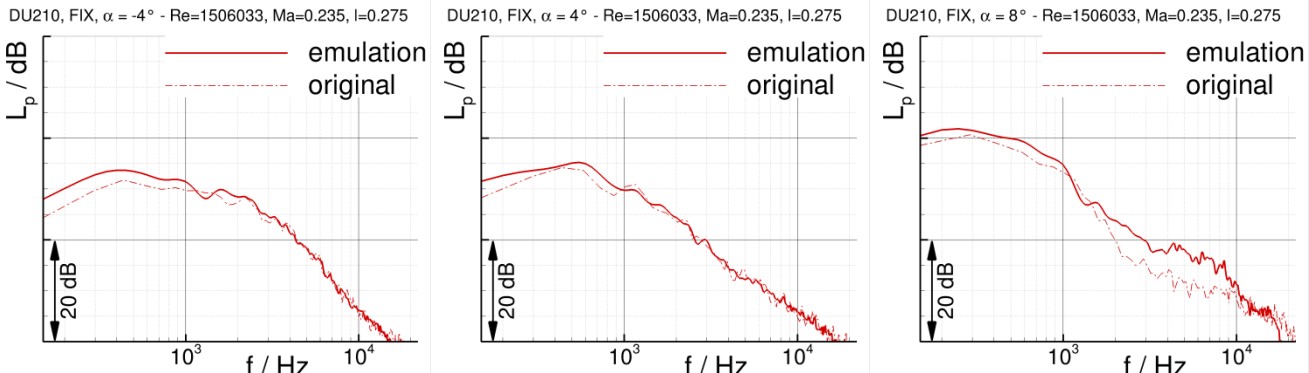

**Figure 60: Comparison of emulated spectra according to eq.3 with $n = m = 2$ and originally simulated spectra at the test condition with fixed transition (FIX) for the airfoil DU210 at $\alpha = -4°, 4°, 8°$.**

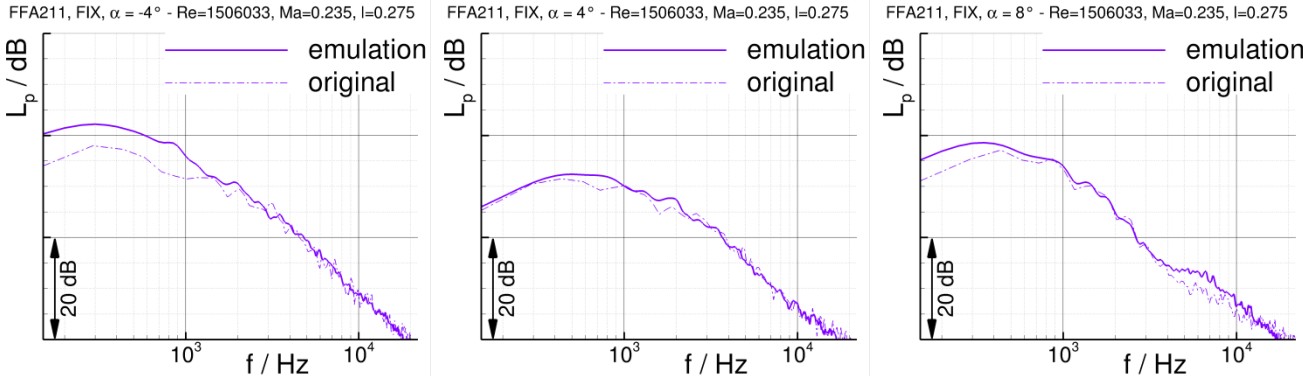

**Figure 61: Comparison of emulated spectra according to eq.3 with $n = m = 2$ and originally simulated spectra at the test condition with fixed transition (FIX) for the airfoil FFA211 at $\alpha = -4°, 4°, 8°$.**

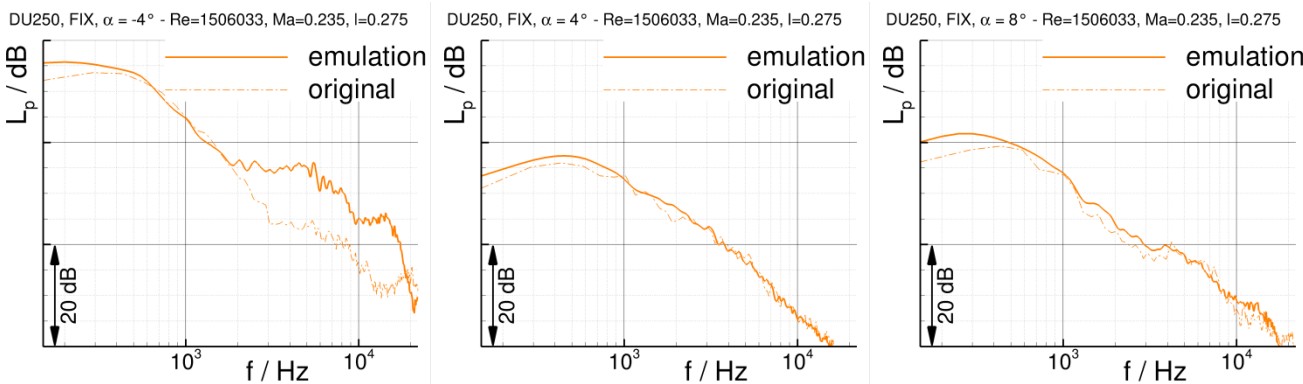

**Figure 62: Comparison of emulated spectra according to eq.3 with $n = m = 2$ and originally simulated spectra at the test condition with fixed transition (FIX) for the airfoil DU250 at $\alpha = -4°, 4°, 8°$.**



At negative angles of attack and low Reynolds number, the airfoils with 24% or 25% relative thickness show a recirculation area near the trailing edge at the pressure side—even in RANS mode. This leads to a reduced accuracy of the two associated spectra at high frequencies. At higher angles of attack the flow is fully attached. In total, the proposed model provides reasonable results at a moderate computational invest.

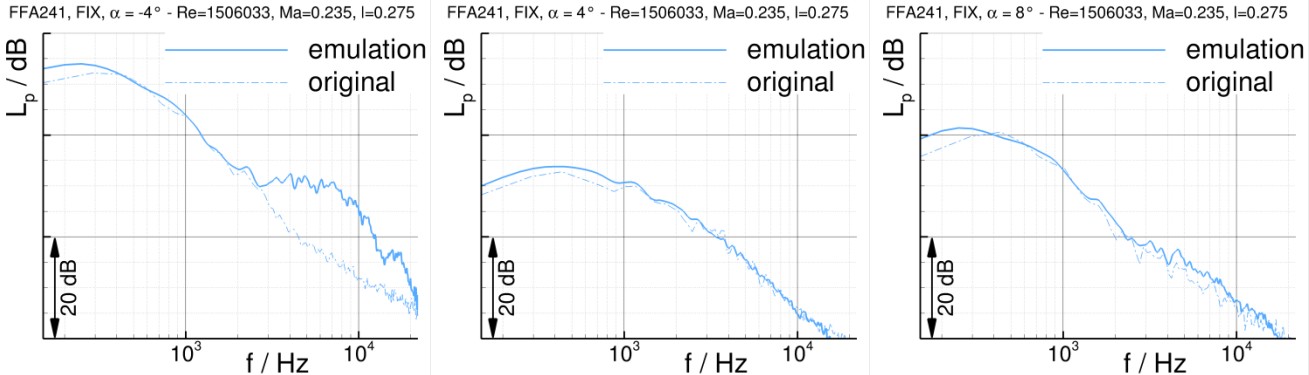

**Figure 63: Comparison of emulated spectra according to eq.3 with $n = m = 2$ and originally simulated spectra at the test condition with fixed transition (FIX) for the airfoil FFA241 at $\alpha = -4°, 4°, 8°$.**

## 3.3.2 Overall sound pressure levels

The OASPL of the simulation results at the test condition (see Table 5) allows for a direct comparison of the airfoils. As expected, the airfoils with a relative thickness of 18% show the lowest values of OASPL. With growing inclination of the airfoil, the emitted sound rises due to the increase of turbulence in the boundary layer. The airfoils with a relative thickness of 24% and above show high OASPL at negative angles of attack. With increasing inclination the emitted sound is temporary reduced, but at higher angles of attack the OASPL rises, again. The investigated airfoils with about 21% relative thickness show an intermediate behavior. At negative angles of attack a highly turbulent zone at the pressure side near the trailing edge causes a higher noise emission. The least noise is emitted at angles of attack near 0°. With rising inclination, the OASPL is rising again. This is illustrated in Figure 64. The OASPL in the figures is reduced to a frequency range of 0.125 kHz to 12.5 kHz. The simulations with partially laminar boundary layer expectedly show a reduced noise production in comparison to the fully turbulent boundary layer. The low noise airfoil LN118 effectively generates the least sound in this 2D CAA.





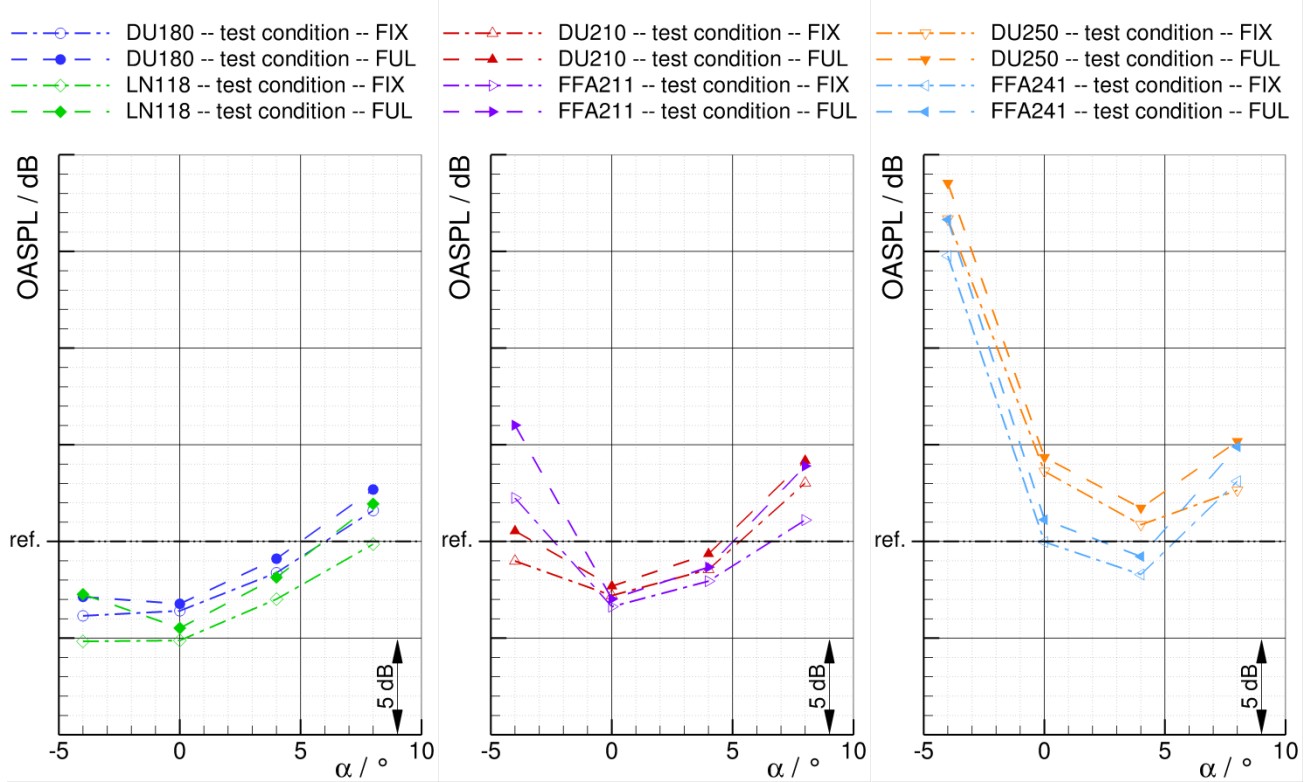

**Figure 64: Change of the overall sound pressure level OASPL with the angle of attack for the acoustically investigated airfoils at the simulated test conditions with fixed transition (FIX) and fully turbulent boundary layer (FUL). The frequency range of the plotted OASPL is reduced to $f = 0,125 \ldots 12,5\text{kHz}$.**

### 3.3.3 Directivity functions

From the microphone records a directivity function is determined. In Figure 65 the above emphasized aeroacoustic characteristic concerning the thickness of airfoils as well as the effect of angle of attack is replicated. The CAA includes both, convective amplification and refraction and diffraction effects, respectively. In addition, the investigated airfoils all are asymmetric. Thus, the simulated directivity functions show clear inclination due to the asymmetry and the angle of attack. Generally, less noise is emitted upstream and downstream, while the directions of 30°…150° and 210°…330° mark the main lobes of the directivity characteristic of TBL-TEN. The OASPL in Figure 65 contains all detected frequencies from the measurement.



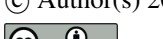

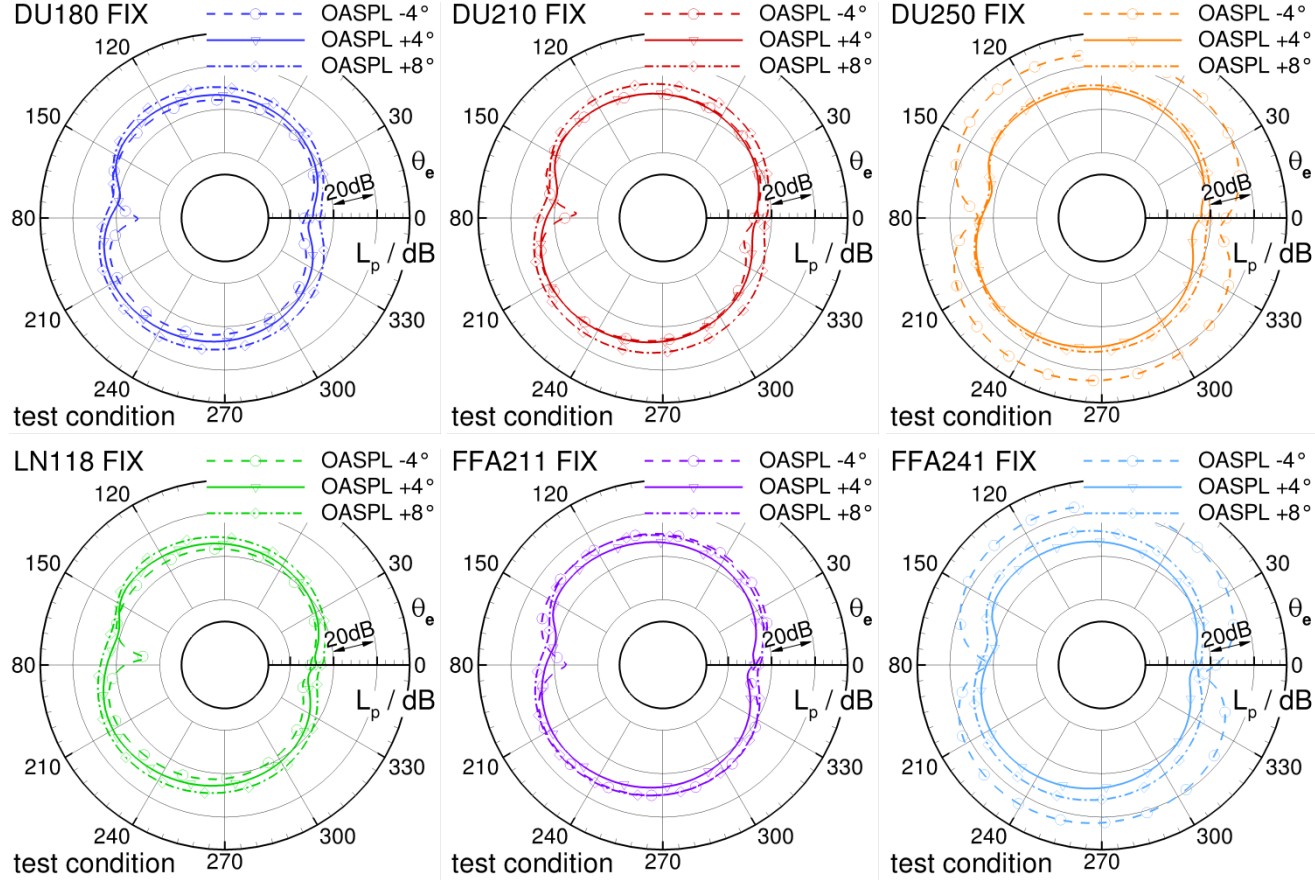

**Figure 65: Directivity functions $D_\Theta$ for the acoustically investigated airfoils at the simulated test conditions with fixed transition (FIX). The plotted OASPL comprises the full frequency range of the evaluated signal.**

## 4 Summary and Discussion

An engineering approach for generating 360° aerodynamic airfoil polars for arbitrary airfoil shapes at moderate cost is presented. The proposed method is validated against experimental data of two airfoils from the TU Delft. It could be shown through the comparison of computed and measured data that the applied methods are superior to the still widely used

5    approximation methods and more simple methods such as XFOIL. Especially the exact prediction of the minimum and maximum lift coefficient as well as capturing the different drag coefficient values for positive and negative angles of attack is greatly improved by the proposed method at still reasonable costs. Because neither one of the methods was superior for all angles of attack, the advantage of each method was exploited through combining the results of both methods via a kriging interpolation model. The approach was subsequently applied to ten different airfoils. The results are published in the current

10    paper and may serve the reader for further comparison with data from other sources. On the contrary the comparison between the simulated and measured drag coefficients around -90° and 90° angle of attack (as well as the moment





coefficients) reveals that the proposed approach is not optimal within this region. A more sophisticated approach, e.g. a computation using IDDES as proposed by N. N. Sørensen during the WESC Conference in Copenhagen is needed in order to determine the exact value for these highly separated flow regions. For the current publication this kind of calculation has not been considered because it is at least two orders of magnitude more expensive than the proposed approach. To the author's

opinion a multi-fidelity approach consisting of 2D RANS and more sophisticated computations such as IDDES show a great potential in providing exact results for the complete polar at still justifiable costs.

In the second part of the paper an aeroacoustic method predicting frequency spectra at a target condition was proposed. It uses pre-calculated anchor conditions and provides reasonable results at moderate costs. The bilinear interpolation of spectra is based on the assumption that only velocity and viscosity effects affect generation of turbulent boundary-layer trailing-edge

noise (TBL-TEN). The presented level shift is influenced by two model exponents $n$ and $m$. The results show a good agreement between narrow band spectra that are simulated and those that were emulated by the new model with $n = m = 2$. In a scheduled contribution to the AIAA/CEAS Aeroacoustic conference in 2018 the exponents will be individually estimated for each airfoil and angle of attack. The estimation will be performed by regression of level differences at same Reynolds (for $n$) and same Mach number (for $m$). As shown above, the exponents tend to $2 < n < 4$ and $0 < m < 2$. One

should expect a typical scaling of $n + m + 1 = 5$ for the overall sound pressure level (OASPL) and $n + m = 4$ for the narrowband spectra.

The overall sound pressure level of airfoils with medium relative thickness behaves intermediate in comparison to those with higher or lower relative thickness. Thus, the blending of angles of attack and different airfoil geometry seems feasible to meet arbitrary airfoils geometry and flow conditions along the rotor blade.

The quality of the results is dependent on robust RANS simulations. Some combinations of Mach number, Reynolds number, airfoil and angle of attack reveal agile and sensitive behaviour of the flow. This is expected at high angles of attack were aerodynamic stall is to be due. On the other hand, airfoils with high relative thickness may show highly turbulent areas near the trailing edge at the pressure side of the airfoils at low or negative angles of attack. To keep the computational effort as little as possible, these effects are accepted as long as they reflect true physics. The proposed method will provide a good

estimation of the relative levels in a direct comparison of the emitted noise levels of two different rotors, even if single deviating anchor conditions may distort certain emulation spectra. An exact prediction in absolute levels is not the objective, yet.

The emulated spectra at objective conditions will soon be processed to plot the OASPL and the associated directivity function in comparison to the simulated ones. This step is a further stage of the verification of the method. During this

ongoing work, the generated data will finally be used to estimate the sound emission at an arbitrary observer position, caused by the respective wind turbine, like illustrated in Figure 7.



**Acknoledgements**

The first author expresses his gratitude towards W.A. Timmer from TU Delft for providing the experimental data set of the two airfoils DU-93-W-210 and DU-97-W-300.

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
