# Peer review of "Numerical airfoil catalogue including $360^{\circ}$ airfoil polars and aeroacoustic footprints"

_Wind Energy Science, 2017_

## Referee Comment (RC1) · G. Cortina (Referee) · 9 Jan 2018

**Specific questions and comments to address for the paper entitled *'Numerical airfoil catalogue including 360°* *airfoil polars and aeroacoustic footprints'***

*Summary: In this article, in a first step the authors present a methodology for generating 360° airfoil polars and aeroacoustic characteristics by means of CFD and CAA. Moreover, results are presented for ten different airfoils and the results obtained by the aforementioned procedure are validated against experimental data of well-known airfoils. Moreover, in this work it is shown that by combining two data-sets obtained by different turbulent models and meshing typologies results show good approximation with experimental data. In a second step the authors provide the aeroacoustic characteristics for a wide operation range. Specifically, the corresponding overall sound pressure level for five angles of attack for six different airfoils is presented and the difference between a fully turbulent computation and simulations with fixed transition is assessed.*

*Below, my comments are categorized as being either Major concerns or Minor concerns, where the former relate to conceptual technical critiques while the latter relate to grammatical/spelling errors. My recommendation is* **major revisions**.

**A: Major Concerns**

1. Abstract: In the abstract, the authors have described what they have done, but they do not explain what the main purpose of this paper is (i.e. to use the data for real wind turbine blades). Also a sentence is required explaining the most relevant results of the second section of the paper (regarding the aeroacoustic footprints).

2. Introduction, p1, line 28: Can you reference the following statement with an existing publication? *'...aerodynamic coefficients for the use in wind energy applications need to be provided for the complete range of angles of attack (from 0° to 360°)'*.

3. Introduction, p2, lines 6-7: Can you include a reference to the following statement? *'These approximation methods are mostly based on a combination of harmonic functions and require certain empirical input parameters'*

4. Tools and Methodology, p3, Figure 1: What transition location means?

5. Tools and Methodology, p6: Why do you chose a speed of 40 m/s? You should reference the selection of the inflow velocity, or discuss your arguments for choosing this velocity. Maybe you can compute the relative inflow velocity of a given wind turbine blade, given a standard inflow velocity to the rotor (for example 10 m/s) and knowing the rotational speed of the rotor through the power curve of a given wind turbine. Then you can also justify the selection of the Reynolds number for the analysis. You might include this analysis into your paper.

6. Tools and Methodology, p6, line 14: How did you select the turbulence intensity?

7. Tools and Methodology, p10, line 31: In lines 15-19 you state that you perform simulations for four different combinations of Mach and Reynolds numbers, instead in line 31 of page 10, you state that you perform five different combinations as illustrated in Table 5. Please correct the statements from lines 15-19.

8. Tools and Methodology, p11, line 7: Why did you only consider airfoils with relative thickness smaller or equal than 25%?
* * *
**B: Minor Concerns**

1. Abstract: Please, spell out CFD, CAA and SST.

2. Introduction, p1, line 21: Can you spell out the DLR project RoDeO?

3. Introduction, p1, line 23: Change *'has'* by *'have'*.

4. Introduction, p1, line 25: Change *'Blade Element Method (BEM)'* by *'Blade Element Theory (BET)'*.

5. Introduction, p1, line 25: Remove *'method'*.

6. Introduction, p2, line 5: Add a comma after *'methods'*.

7. Introduction, p2, line 5: Do you mean *'exist'* instead of *'exit'*?

8. Introduction, p2, line 18-19: Sudden change between topics. Can you include a sentence that connects the previous statements with the paragraph that starts in line 19? For example, you can state: *'Another problem encountered in full-size wind turbines is...'*.

9. Introduction, p2, line 26: Spell out BPM model please.

10. Introduction, p2, line 28: Spell out TNO model please.

11. Introduction, p2, line 29: Change *'In total'* for another connector, such for example *'generally'* or *'commonly'*.

12. Introduction, p2, line 29-30: I don't understand what do you mean by stating that a separate validation is often lacking in detail. Do you want to say that a separate validation should be required? Rewrite this sentence please.

13. Introduction, p2, line 31-34: Use the same verbal tense for the entire paragraph.

14. Tools and Methodology, p3, line 3: Rewrite the first sentence and explain what is the framework. For example, you can start this fist paragraph as: *'A chart illustrating the framework used to generate aerodynamic polars is shown in Figure 1.'*

15. Tools and Methodology, p3, line 6: Spell out TAU.

16. Tools and Methodology, p3, line 9: What LILO and COCO mean?

17. Tools and Methodology, p3, line 13: What the program POT means? Can you spell out this program name?

18. Tools and Methodology, p3, Figure 1: Change the caption of the figure to be more explicit. You can write something like: *'Chart describing the framework for generating aerodynamic airfoil polar tables'*.

19. Tools and Methodology, p4, last 3 paragraphs: Combine the last three paragraphs of subsection 2.2 in one paragraph.

20. Tools and Methodology, p6, line 1: What do you mean by *'The velocity an airfoil on a wind turbine*? Maybe you want to say that the relative velocity of a wind turbine blade depends on the radial position, right?

21. Tools and Methodology, p6, line 2: There is no relationship between the second sentence of line 2 page 6 and the sentence that starts in line 3 with *'Nevertheless*. Paragraphs you can rewrite this.

22. Tools and Methodology, p6, line 7: Do you mean from -180 to 180. If not, why 170? And, what is the difference between the first sampling rate of 2 degrees and the second sampling rate also set as 2 degrees? Which are the other regions where the sampling rate is equal to 5 degrees?

23. Tools and Methodology, p6, Table 2: Can you set the exponents as $10^6$ instead of e6?

24. Tools and Methodology, p7, line 3: Can you cite any other work from Gerrit Heilers instead of saying that he programmed a set of python scripts? Otherwise omit the reference to the author's name.

25. Tools and Methodology, p7, line 8-9: You might omit the following sentence: *'If one or more computations or even the whole DOE crash due to convergence, other software or hardware errors various techniques for restarting the computations are available'*, since it is not relevant information by the reader.

26. Tools and Methodology, p8, line 10: You are missing the subject of the sentence in the sentence that starts with *'Latter'*, and, can you clarify what is set to 2 and why?

27. Tools and Methodology, p8, line 26: Change $c_{l,max}$ to $C_{l,max}$.

28. Tools and Methodology, p9, lines 15-19: Rewrite the paragraph that comprises lines 15 to 19 for a better understanding of the procedure. You can rewrite this paragraph as follows: *'Unlike the aerodynamic analysis, where only one representative Mach- and Reynolds number was chosen to determine the aerodynamic coefficients, for the aeroacoustic analysis five different combinations of Mach- and Reynolds numbers are selected. This is because from the aeroacoustic point of view, the flow velocity has a greater impact on the emitted sound than its viscosity.'*

29. Tools and Methodology, p11, line 2: Change *'Radius'* by *'radius'*.

30. Tools and Methodology, p12, caption of Figure 6: By writing $f_{max} = 10...60$ kHz, do you mean maximum frequencies from 10 to 60 kHz? Can you clarify this statement please?

31. Tools and Methodology, p12, line 7: Change the dashes that enclose *'spectrally resolved'*, or you can include a space before and after the dashes.

32. Tools and Methodology, p12, line 20: Change the dashes that enclose *'in motion and regarding the directivity function towards the observer'*, or remove them.

33. Tools and Methodology, p13, line 12: Remove *'Aeroacoustic Results)'*.

34. Tools and Methodology, p13, line 17-18: Write eq.1 or simply equations 1 and 2. Same applies for equation 3 in line 18.

35. Tools and Methodology, p13, line 19: Change the dashes that enclose *'in the guise of Power Spectral Densitiy (PSD)'*, or you can include a space before and after the dashes.

36. Tools and Methodology, p14, Figure 9: Change the labels of Figure 9 to $L_p$ (dB) and $f$ (Hz), and include the values of the y-axis.

37. Tools and Methodology, p14, Caption Figure 9: Change *'...test condition at 4° angle of attack...'* for *'...test condition for an angle of attack equal to 4°...'*

38. Aerodynamic and Aeroacoustic Results, p16, Figures 10, 12 and 13: Be consistent with the plot labels and the text. If in the text you use $C_l$ use the same nomenclature for the plot labels instead of $CLift$.

39. Aerodynamic and Aeroacoustic Results, p16, line 3: Change $C_{lmin}$ by $C_{l,min}$.

40. Aerodynamic and Aeroacoustic Results, p17, line 6: Space needed between *than* and $-16°$.

41. Aerodynamic and Aeroacoustic Results, p17, Figure 14 and 15: Be consistent between the labels of the figure and the text.

42. Aerodynamic and Aeroacoustic Results, p18, line 2: Change $C_{lmax}$ by $C_{l,max}$. The same aplies for the rest of the text.

43. Aerodynamic and Aeroacoustic Results, p19, line 21: Change 1e6 to $10^6$.

44. Aerodynamic and Aeroacoustic Results, p22, line 10: Space needed between *'Method 2.'* and *At*.

45. Aerodynamic and Aeroacoustic Results, p22, line 21: Include Figures from 26 to 55 in an Annex Section.

46. Aerodynamic and Aeroacoustic Results, p26, line 4: Remove *'Processing of the aeroacoustic data)'*.

47. Aerodynamic and Aeroacoustic Results, p26, line 5: Remove *'Spatial results)'*.

48. Aerodynamic and Aeroacoustic Results, p26, line 7: Remove *'Overall sound pressure levels)'*.

49. Aerodynamic and Aeroacoustic Results, p26, line 16-17: Remove *'Directivity functions)'*.

50. Aerodynamic and Aeroacoustic Results, p27, Figures 58 to 63: Change the labels of Figures to $L_p$ (dB) and $f$ (Hz), and include the values of the y-axis.

51. Aerodynamic and Aeroacoustic Results, p27, line 9: Include a space between the numbers and the units, $l_c = 0.101$ m.

52. Aerodynamic and Aeroacoustic Results, p29, line 2: Change the dash before the text *'even in RANS mode'*, or you can include a space before the dash.

53. Aerodynamic and Aeroacoustic Results, p30, Figure 64: Change the labels to OASPL (dB) and $\alpha$ (°), and include the values of the y-axis.

54. Aerodynamic and Aeroacoustic Results, p30, caption of Figure 64: By writing $f = 0,125...12,5$ kHz, do you mean frequencies from 0,125 to 12,5 kHz? Can you clarify this statement please?

55. Summary and Discussion, p32, line 2: Can you include a reference to the approach proposed by Sørensen?

---

## Referee Comment (RC2) · Anonymous Referee #2 · 26 Jan 2018

The manuscript submitted by the authors is in my opinion more an internal progress report than a scientific journal article. The manuscript includes 65 figures presenting results of numerical simulations in form of aerodynamic polars and acoustic spectra without critical analysis of the results. In my opinion, the manuscript lacks a clear objective and does not contribute new insights to our current state of knowledge on the aerodynamics or aeroacoustics of wind turbine blades. It merely serves as a database.

If it is the authors' sole purpose to provide a database of numerically obtained aerodynamic and aeroacoustic properties of wind turbine blades, I believe that they should still provide further justification on:

- their choice of Re and free stream velocity (why 40m/s?). How do these numbers

relate to relevant conditions encountered by wind turbines?

- the choice of turbulence intensity of 0.001. This does not seem to be a relevant level of turbulence intensity in the atmospheric boundary layer.

- why they consider a range of angles of attack of 360

- the use of RANS. As the authors state multiple times, their RANS calculations "suffer from severe converge problems" and do not capture the separation behaviour. While I understand the need for low order models to predict the aerodynamic and aeroacoustic behaviour of wind turbines blades for design and flow control purposes, the authors do not provide a sufficient argumentation how their RANS calculations can contributed to the derivation of improved low order models.

- the validity of their fusion approach and how they decide on when to select the results of one method in favour of the other. The authors validated their three numerical methods based on a single incomplete data set (p19: the exact procedure for obtaining the experimental values is unknown). They conclude that neither of the numerical approaches was superior to all the angles of attack and propose to combine results of the both method without justification nor validation of the result.

- the relevance of using 2D steady simulations to provide insight into an inherently unsteady 3D problem

Furthermore:

- the authors use a lot of abbreviations which they do not explain

- figures should be cited in the order in which they appear

---

## Author Comment (AC1) · 16 Feb 2018

The authors would like to thank the first Referee for his close inspection of the manuscript, his constructional comments and clear and detailed proposal of corrections. Almost all corrections have been included in the revised manuscript. For discussion of the major concerns please check the suplemented file.

Please also note the supplement to this comment:
https://www.wind-energ-sci-discuss.net/wes-2017-51/wes-2017-51-AC1-supplement.pdf

**Supplement:**

**Major Concerns**

1. Abstract: p1, line 19 – 21
2. See manuscript
3. See manuscript
4. Transition from laminar to turbulent flow
5. If one applies Buckingham's π-theorem, it can be deduced that the Reynolds number is the critical characteristic number for this problem. As now stated in the paper more clearly the final goal of generating the polar tables is the design of a wind turbine blade including further improvements again based on the polar tables. The rotor to be designed will have a radius of 20m. Choosing a typical rotational speed (such that the tip speed will be around 80 m/s), a preliminary design including the chord length distribution along the radius can be approximated. From this an approximate Reynolds number distribution along the radius can be approximated. This Reynolds number will be in the order of $1*10^6$. For a pure aerodynamic analysis the freestream velocity can be chosen almost arbitrarily. For technical reasons (small velocities in a compressible solver will result in a stiff system of equations) a very small inflow velocity is not recommendable. Also choosing the tip velocity (or even above) as the inflow velocity is not recommendable since Mach number effects will start to become relevant. Therefore the intermediate velocity of 40 m/s has been chosen as a representative inflow speed. Inflow speeds of 30 m/s or 50 m/s would give the same results.
6. The main reason for choosing this turbulent intensity is not the atmospheric turbulence but the comparison with the experimental data.
7. A further sentence was added for improvement
8. A further comment was added to stress that only relatively thin airfoils suit for the outer 20% of the rotor

**Minor Concerns**

36. Labels changed, but no values for the vertical axis added. The relative measure must do.
50. Ditto
53. Ditto

---

## Author Comment (AC2) · 16 Feb 2018

Response to referee comments

General:

The purpose of the present paper is the publication of a database for publicly available state of the art airfoils containing the relevant aerodynamic **AND** aeroacoustic information for the design of a wind turbine blade. This database enables the reader to choose adequate airfoils during the design process for an aerodynamically and aeroacoustically optimal rotor. To the author's knowledge such a database currently does not exist publicly available. **Furthermore** the chosen approach is described in detail, the aerodynamic methods are validated against experimental data and therefore critically analyzed. As the first author states in the paper, an engineering approach is selected for the computation of the aerodynamic coefficients rather than the highest available fidelity approach. The reason for this is the trade-off between accuracy and efficiency. For achieving more accurate aerodynamic coefficients a more sophisticated 3D approach is necessary, but this results in computational times which are at least 1-2 orders of magnitude higher. On the contrary, the results presented in the paper are much more accurate than what can be achieved with empirical functions as described in Skrzypiński et al. Moreover the proposed method is free of empiricism.

The answers to the referee's responses are as follows (ordered from top to bottom):

1. If one applies Buckingham's π-theorem, it can be deduced that the Reynolds number is the critical characteristic number for this problem. As now stated in the paper more clearly the final goal of generating the polar tables is the design of a wind turbine blade including further improvements again based on the polar tables. The rotor to be designed will have a radius of 20m. Choosing a typical rotational speed (such that the tip speed will be around 80 m/s), a preliminary design including the chord length distribution along the radius can be approximated. From this an approximate Reynolds number distribution along the radius can be approximated. This Reynolds number will be in the order of $1*10^6$. For a pure aerodynamic analysis the freestream velocity can be chosen almost arbitrarily. For technical reasons (small velocities in a compressible solver will result in a stiff system of equations) a very small inflow velocity is not recommendable. Also choosing the tip velocity (or even above) as the inflow velocity is not recommendable since Mach number effects will start to become relevant. Therefore the intermediate velocity of 40 m/s has been chosen as a representative inflow speed. Inflow speeds of 30 m/s or 50 m/s would give the same results.
2. The main reason for choosing this turbulent intensity is not the atmospheric turbulence but the comparison with the experimental data.
3. If all operational conditions according to IEC regulations should be considered it is necessary to be able to provide the aerodynamic coefficients of the airfoils integrated in the wind turbine blade for angles of attack from 0° to 360°.
4. The authors are aware of the convergence problem and have therefore analyzed the results closely according to the convergence measures (density residual, Cl-amplitude). Non-converged solutions are excluded from the final polar or their impact is smoothed according to the

regressive interpolation. The improved polars (improved means better results than based on empirical functions) are subsequently used in a comprehensive rotor code based on the BET for analyzing the various operational conditions.

5.  The referee comment "The authors validated their three numerical methods based on a single incomplete data set" is unclear to the authors. The experimental data set which is used for comparison is one of the most complete publicly available data sets. The method how the data sets are combined is described in section 2.5. The general procedure is referenced to further publications and is a valid procedure as can be seen in case the referenced publications are studied closely. The fact that neither numerical approach was superior to the others for **ALL** angles of attack is the reason why the (intelligently) combined dataset is superior to the single datasets.

6.  See general comment

7.  Has been adjusted according to comments from first referee